# Quantitative LLM Judges

## Abstract

LLM-as-a-judge is a framework where a large language model (LLM) evaluates the output of another LLM. While LLMs excel at producing qualitative textual evaluations, they often struggle to predict human preferences and numeric scores. We propose *quantitative LLM judges*, which align evaluation scores of LLM judges to humans in a given domain using regression models. These models are trained to improve the score of the original judge using its rationale and score. We present four quantitative judges for different types of absolute and relative feedback, which showcases the *generality* and *versatility* of our framework. Our framework does not require fine-tuning and can be applied when human feedback is limited, which is expected in practice. We validate our claims on four datasets. Our experiments show that quantitative judges consistently outperform base judges on optimized metrics and are competitive with specialized LLM judges, a larger off-the-shelf model used as a judge, and fine-tuning by TRACT.

## 1 Introduction

Measuring the quality of generated natural language presents several challenges due to the diverse range of generation methods and evaluation criteria (Gehrmann et al., 2023). The *LLM-as-a-judge* paradigm has recently emerged as a compelling approach to these evaluation challenges. By leveraging the reasoning capabilities of *large language models (LLMs)*, this approach can provide more nuanced assessments that correlate better with human judgments across diverse tasks (Gu et al., 2025). LLM judges output both rationales and numeric scores, thus combining the comprehensiveness of human evaluation with the scalability of automated metrics.

Recent studies highlight issues with LLM judges such as low alignment with human scores, miscalibration, score compression, high variance, prompt sensitivity, and leniency bias (Thakur et al., 2025; Wei et al., 2025). To address these issues, fine-tuning and reward models have been recently proposed (Chiang et al., 2025; Lukasik et al., 2025; Ankner et al., 2025). The main challenge with applying these approaches in practice is that they require access to model weights or training another complex model. On the other hand, specialized LLM judges, such as Prometheus (Kim et al., 2024), are fine-tuned open-weight models on gold-standard textual evaluations and human scores. These judges are applied through prompting and may perform worse than larger pre-trained models. As a result, the most common LLM judges are larger off-the-shelf models (Bavaresco et al., 2024; Huang et al., 2024).

In this work, we decouple qualitative rationales from quantitative scores in a generic way. This allows us to use any LLM to produce high-quality rationales while numeric scores are predicted by classic machine learning models that are robust and loved by practitioners. This perspective is supported by prior works in interpretability and probing, which show that when model representations encode information relevant for downstream tasks, simple linear decoders can recover this information effectively (Alain & Bengio, 2017; Hewitt & Manning, 2019; Hupkes et al., 2018; Belinkov, 2022).

Building on this insight, we present *quantitative judges*, a framework that enhances the original base judge to predict more accurate numeric scores. We introduce four different quantitative judges for absolute rating and relative preference prediction. Each judge has two stages: in the *qualitative stage*, a frozen LLM judge generates a rationale and initial score, and in the *quantitative stage*, these outputs are used to predict a better score. Our design is general, lightweight, efficient, and applies to any base LLM judge. Specifically:

1. **General.** Our judges predict human scores from the embedding of the base judge's rationale and its score. The predictor is a *generalized linear model (GLM)* (McCullagh & Nelder, 1989) trained on human scores in the domain. The judges can be applied to absolute rating prediction, such as regression and classification, or relative preference prediction. We showcase the versatility of our framework by proposing four quantitative judges.

2. **Statistically efficient.** Our judges are based on GLMs, which can be learned from limited data. This is expected in most applications of our work. They are also biased to the base judge's score or a distribution of it. This allows, at least in principle, learning of at least as good judges as the base judge (Appendix C.2).

3. **Computationally efficient.** Learning of classic machine learning models on the top of frozen LLM embeddings is more computationally efficient than fine-tuning and can also outperform it.

We comprehensively evaluate quantitative judges on absolute rating and relative preference prediction datasets. They consistently outperform base judges on optimized metrics; at the expense of potentially worsening other metrics. In most of our benchmarks, they outperform or are comparable to specialized LLM judges, a larger off-the-shelf model used as a judge, and fine-tuning by TRACT (Chiang et al., 2025). This is despite the fact that the quantitative judges have only a few thousand parameters, and are learned from a few hundred ratings or pairwise comparisons. These results are complemented with additional studies on key components of our design. We conclude that quantitative judges are a practical, lightweight, and effective approach for improving LLM-based evaluation.

## 2 Background

**LLM-as-a-judge paradigm.** LLMs have increasingly been adopted as evaluators of model outputs, a paradigm now widely referred to as *LLM-as-a-judge* (Kim et al., 2024; Gu et al., 2025). By leveraging natural language reasoning, LLM judges can produce free-form textual rationales, numeric scores, or pairwise preferences, which serves as a scalable and reproducible alternative to human evaluation (Kim et al., 2024). Empirical studies show that such judges often correlate more strongly with human judgments than traditional heuristic metrics, particularly for open-ended generation tasks (Kim et al., 2024; Gu et al., 2025). At the same time, multiple works demonstrate that the numeric scores produced by LLM judges are frequently miscalibrated, compressed, and highly sensitive to prompt formulation, which gives rise to inconsistent alignment with human assessments across datasets and domains (Chiang & Lee, 2023; Gehrmann et al., 2023).

**Fine-tuning and self-improvement.** LLM judge's reliability can be improved by fine-tuning it. Prometheus (Kim et al., 2024) and JudgeLM (Zhu et al., 2025) are fine-tuned open-weight models on public gold-standard textual evaluations and human scores. These models can be used for both absolute rating and relative preference prediction, and we experiment with them in Section 5. LLM judges have been trained on various instruction-response datasets, such as Alpaca52k (Bommasani et al., 2021), LMSYS-Chat (Zheng et al., 2024), ToxicChat (Lin et al., 2023), and LLMEval (Zhang et al., 2024), often using feedback from stronger proprietary models. Self-improvement through rationalization or critique generation, where judges generate chains of thought, critiques, or preference signals that are then used for optimization, was explored by Yuan et al. (2024) and Trivedi et al. (2024). *Regression-aware fine-tuning (RAFT)* (Lukasik et al., 2025) and *two-stage regression-aware fine-tuning with CoT (TRACT)* (Chiang et al., 2025) can be used to fine-tune judges to a particular domain. We compare to TRACT in Section 5.

**Decoupled evaluation and post-hoc modeling.** Our work is motivated by the observation that, despite poor numeric calibration, LLM judges often produce high-quality qualitative rationales that encode rich, domain-specific information relevant to human judgment. This observation is consistent with prior work in interpretability and representation learning, which shows that frozen model representations contain linearly recoverable signals for downstream tasks (Alain & Bengio, 2017; Belinkov, 2022). Rather than fine-tuning the judge itself, we explicitly decouple qualitative reasoning from quantitative scoring. We fix rationales and scores generated by a base LLM judge and train a lightweight quantitative model to align these signals

with human annotations, using as few as 100–200 labeled examples. Despite its simplicity, this approach consistently outperforms substantially larger and heavily fine-tuned judge models across datasets, while exhibiting significantly greater statistical efficiency and stability under limited feedback regimes. This reframing treats LLM judges as feature generators rather than end-to-end evaluators, yielding a practical and robust alternative to direct judge fine-tuning (Gu et al., 2025).

## 3 Setting

We study two types of LLM judges: for evaluating a single response and comparing two responses.

**Absolute LLM judge.** The absolute judge evaluates a single LLM response. The evaluation can have various forms: text, score, or both. The score can evaluate various aspects of the response, such as coherence, correctness, factual consistency, relevance, and adherence to task-specific guidelines. In this work, we assume that the judge generates both a rationale and its score. Specifically, let $(x, y)$ be a prompt-response pair from a judged LLM. The *absolute judge* maps $(x, y)$ to $(e, b)$, where $e$ is a *rationale* that judges $y$ given $x$ and $b \in \mathbb{R}$ is the associated *absolute score*.

The primary advantage of an absolute judge is its consistency and standardization in scoring across different responses. However, it may require extensive prompt engineering or fine-tuning to align with human scores (Kadavath et al., 2022). The relative judge, which we introduce next, mitigates it by making a direct comparison. However, it may introduce biases such as sensitivity to the order of the compared responses (Jeong et al., 2024).

**Relative LLM judge.** The relative judge compares two or more LLM responses, and ranks them or selects the best one. It is commonly used in ranking-based assessments, preference modeling, and pairwise comparisons. Formally, let $(x, y_1, y_2)$ be a prompt-responses tuple from two judged LLMs. The *relative judge* maps $(x, y_1, y_2)$ to $(e, b)$, where $e$ is the *rationale* that judges $y_1$ and $y_2$ given $x$, and $b \in \{0, 1\}$ is the associated *relative preference score*. When $b = 1$, the judge prefers the first response $y_1$; otherwise it prefers the second response $y_2$. We consider pairwise comparisons to simplify exposition and discuss an extension to multiple responses in Appendix C.1.

## 4 Quantitative LLM Judges

This work is motivated by the observation that the scores of pre-trained LLM judges (Section 3) are not calibrated to any given domain, simply because they are trained on general world knowledge. To obtain better scores, we learn to predict them from rationales of existing judges and human scores in the domain. Our predictors are *generalized linear models (GLMs)* (McCullagh & Nelder, 1989), which extend linear models to non-linear functions while inheriting their efficiency. We get a more quantitative judge by using the predicted score and thus call our judges *quantitative*.

We introduce four quantitative judges and each has the following high-level structure. The existing judge is called a *base judge*, and we assume that its evaluation comprises both a rationale and score. We denote its *rationale* by $e$ and its vector *embedding* by $\phi(e) \in \mathbb{R}^d$, where $d$ is the embedding dimension. The embedding can be obtained from the base judge when it is open-weight, as described in Section 5.2, or from another model, as in Appendix B.2. Therefore, our approach can be viewed as black box. We denote the *base judge's score* by $b$. When the base judge assigns probabilities to its scores, we denote them by $p$. At *inference time*, when we judge, a human score is predicted from $\phi(e)$, along with $b$ or $p$. At *training time*, we use *ground-truth human scores* to train the predictor and denote them by $s$. We show architectures of our quantitative judges in Figure 1 and present them next.

### 4.1 Least-Squares Judge

The *least-squares (LS) judge* is an absolute judge that predicts the score of a single response as

$$f(e, b; \theta) = (\phi(e) \oplus b)^\top \theta + c, \tag{1}$$

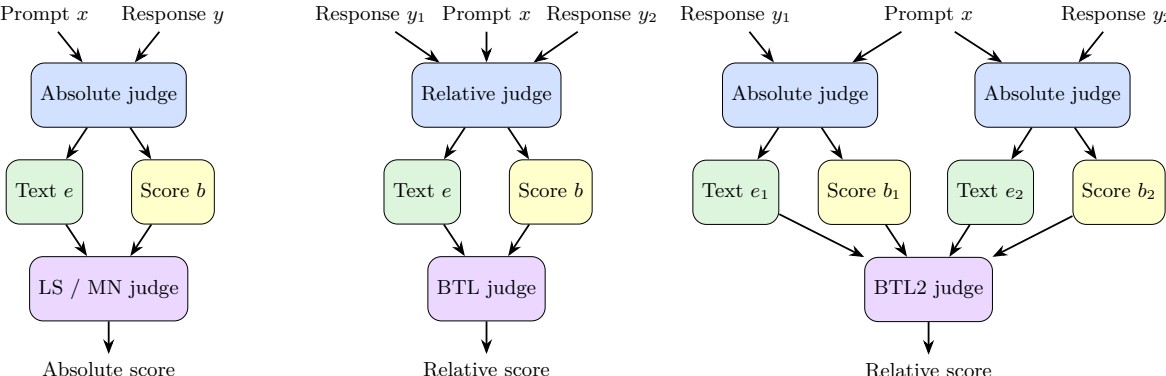

Figure 1: Architectures of the least-squares (LS), multinomial (MN), Bradley-Terry-Luce (BTL), and two-headed BTL (BTL2) judges described in Section 4.

where $\phi(e) \in \mathbb{R}^d$ is the embedding of base judge's rationale, $b \in \mathbb{R}$ is its score, $u \oplus v$ represents a concatenated vector of $u$ and $v$, $\theta \in \mathbb{R}^{d+1}$ is a learned model parameter, and $c$ is a population bias. The population bias plays the same role as the bias term in linear regression. We introduce $b$ so that we can learn at least as good of a judge as the base judge. To see this, note that (1) becomes the base judge's score $b$ when $\theta = \mathbf{0}_d \oplus 1$ and $c = 0$. We justify this design formally in Appendix C.2: under the assumptions of large data ($n \to \infty$) and no regularizarion ($\gamma = 0$), we prove that the learned $\theta$ improves upon the base judge with a high probability.

We learn the judge by minimizing the squared loss, which lends the judge its name. Specifically, we minimize a *regularized squared loss*

$$\mathcal{L}(\theta) = \sum_{t=1}^{n} (f(e_t, b_t; \theta) - s_t)^2 + \gamma \|\theta\|_2^2$$

over $n$ data points, where $e_t$ is the base judge's rationale, $b_t$ is its score, and $s_t$ is the human score in data point $t \in [n]$. The regularization strength $\gamma > 0$ is set automatically by cross-validation.

## 4.2 Multinomial Judge

The *multinomial (MN) judge* is an absolute judge designed for predicting categorical scores, such as on the Likert scale. The Likert score can be viewed as an absolute score or as a ranking among options (Carifio & Perla, 2008). The MN judge is designed for the former. Our relative judges in Sections 4.3 and 4.4 are designed for the latter.

The MN judge predicts the most likely score from a set $\mathcal{S}$. The probability of score $s \in \mathcal{S}$ is $\pi(s \mid e, p; \Theta) =$

$$\frac{\exp[(\phi(e) \oplus \log p_s)^\top \theta_s + c_s]}{\sum_{s \in \mathcal{S}} \exp[(\phi(e) \oplus \log p_s)^\top \theta_s + c_s]}, \tag{2}$$

where $\phi(e) \in \mathbb{R}^d$ is the embedding of base judge's rationale, $p_s$ is the probability that the base judge predicts score $s$, $p = (p_s)_{s \in \mathcal{S}} \in \Delta^{\mathcal{S}}$ is a probability vector, $\theta_s \in \mathbb{R}^{d+1}$ is a learned model parameter for score $s$, and $c_s$ is a population bias towards $s$. We denote all learned parameters by $\Theta = (\theta_s)_{s \in \mathcal{S}}$ and estimate $p_s$ using the next token probability in $e$ (Gu et al., 2025, Section 2.3.3).

Since (2) is equivalent to the probability of outcome $s$ in *multinomial logistic regression* (Murphy, 2012, Chapter 8), the population bias $c_s$ plays the same role. We introduce $p$ so that we can always learn a judge that performs at least as well as the base judge. Specifically, when $\theta_s = \mathbf{0}_d \oplus 1$ and $c_s = 0$ for all $s \in \mathcal{S}$, the predicted probability becomes the base judge's probability because

$$\frac{\exp[\log p_s]}{\sum_{s \in \mathcal{S}} \exp[\log p_s]} = \frac{p_s}{\sum_{s \in \mathcal{S}} p_s} = p_s \,.$$

The last equality holds because $p$ is a probability vector and thus $\sum_{s \in \mathcal{S}} p_s = 1$.

We learn the judge by maximizing the probability of correct score predictions. This is equivalent to minimizing a *regularized cross-entropy loss*

$$\mathcal{L}(\Theta) = -\sum_{t=1}^{n} \log \pi(s_t \mid e_t, p_t; \Theta) + \gamma \|\Theta\|_2^2 \,,$$

where $e_t$ is the base judge's rationale, $p_t$ is a distribution over its scores, and $s_t$ is the human score in data point $t \in [n]$. The regularization strength $\gamma > 0$ is set automatically by cross-validation.

We consider both LS and MN judges because they provide two different perspectives on predicting an absolute score: regression versus classification. The LS judge treats the scores as real numbers and minimizes the squared error. The MN judge treats the scores as discrete choices and maximizes the accuracy of predicting them.

### 4.3 Bradley-Terry-Luce Judge

The *Bradley-Terry-Luce (BTL) judge* is a relative judge that estimates the preference of one response over another from its evaluation by a relative base judge. The judge is motivated by the most popular discrete choice model in human preference modeling (Bradley & Terry, 1952). The probability that the first response is preferred is computed as

$$\pi(e, p; \theta) = \mu\left( \left( \phi(e) \oplus \log \frac{p}{1-p} \right)^\top \theta + c \right) , \tag{3}$$

where $\mu$ denotes a *sigmoid function*, $\phi(e) \in \mathbb{R}^d$ is the embedding of base judge's rationale, $p$ is the probability that the base judge prefers the first response, $\theta \in \mathbb{R}^{d+1}$ is a learned model parameter, and $c$ is a population bias. We estimate $p$ using the next token probability in $e$ (Gu et al., 2025, Section 2.3.3). The first response is preferred when $\pi(e, p; \theta) > 0.5$; otherwise the second one is preferred.

The population bias $c$ plays the same role as the bias term in logistic regression. We introduce $p$ so that we can always learn a judge that performs at least as well as the base judge. Specifically, when $\theta = \mathbf{0}_d \oplus 1$ and $c = 0$, the predicted probability becomes the base judge's probability because

$$\mu\left( \log \frac{p}{1-p} \right) = \frac{1}{1 + \exp\left[ -\log \frac{p}{1-p} \right]} = \frac{1}{1 + \frac{1-p}{p}} = p \,.$$

We learn the judge by maximizing the probability of ranking correctly. We pose this as minimizing a *regularized logistic loss*

$$\mathcal{L}(\theta) = -\sum_{t=1}^{n} [s_t \log \pi(e_t, p_t; \theta) + (1 - s_t) \log(1 - \pi(e_t, p_t; \theta))] + \gamma \|\theta\|_2^2 \,,$$

where $e_t$ is the base judge's rationale, $p_t$ is the probability that the judge prefers the first response, and $s_t \in \{0, 1\}$ is the human score in data point $t \in [n]$. When $s_t = 1$, the human prefers the first response; and when $s_t = 0$, the human prefers the second one. The regularization strength $\gamma > 0$ is set automatically by cross-validation.

### 4.4 Two-Headed BTL Judge

The *two-headed BTL (BTL2) judge* is a BTL judge that estimates the preferred response from two separate absolute evaluations. This builds on the findings that pointwise evaluators tend to be more robust (Jeong et al., 2024), while pairwise evaluators are more susceptible to superficial cues due to inherent biases in LLMs (Wang et al., 2024a; Chiang & Lee, 2023). Our empirical results in Table 2 strongly support this approach.

We instantiate BTL2 within the framework of Section 4.3 as follows. Let $\phi(e_1), \phi(e_2) \in \mathbb{R}^d$ be the embeddings of base judge's rationales $e_1$ and $e_2$, respectively; and $b_1, b_2 > 0$ be the associated scores. We define the probability that the first response is preferred as (3), where $\phi(e) = \phi(e_1) - \phi(e_2)$ and $p = b_1/(b_1 + b_2)$. This representation is motivated by the fact that the difference of the embeddings $\phi(e) = \phi(e_1) - \phi(e_2)$ reflects the difference of word affinities in the two responses. The probability $p$ biases the response towards the base judge. Specifically, when $b_1 > b_2$, $\log \frac{p}{1-p} > 0$ and the first response is preferred; otherwise the second response is preferred. The judge is learned exactly as in Section 4.3. We set $p_t = b_{t,1}/(b_{t,1} + b_{t,2})$, where $b_{t,1}, b_{t,2} > 0$ are the base judge's scores for both responses in data point $t \in [n]$. Note that learning and inference with BTL2 is twice as expensive as with BTL because BTL2 relies on the predictions of two base judges. We extend BTL2 beyond a binary comparison in Appendix C.1.

## 5 Experiments

To validate the performance of our quantitative judges, we comprehensively evaluate them on four tasks: two for absolute rating and two for relative preference prediction. We compare them to their base judges, specialized LLM judges, an order of magnitude larger judge, and regression-aware fine-tuning by TRACT.

### 5.1 Datasets and Metrics

We experiment with 4 datasets. The first two datasets contain human absolute ratings and are widely used in the LLM evaluation literature. The last two datasets contain synthetic relative comparisons, testing robustness to adversarial scenarios and scalability. Our dataset selection ensures that we test on both calibrated human judgments and challenging synthetic ground truth. We provide more details next.

**Summarize from Feedback** (Stiennon et al., 2020) is a human-annotated dataset with summary responses rated on a 7-point Likert scale. We use its axis subset, which contains absolute scores for overall helpfulness, accuracy, coverage, and coherence. We use the overall score in our experiments, train on their validation set (8.59k data points), and test on their test set (6.31k data points).

**HelpSteer2** (Wang et al., 2024b;c) is a dataset of human absolute ratings for instruction-response pairs with correctness, coherence, complexity, verbosity, and overall helpfulness scores on a 5-point Likert scale. We use the overall score in our experiments, train on their training set (20.3k data points), and test on their validation set (1.04k data points).

**Offset Bias** (Park et al., 2024) is a synthetic pairwise preference dataset composed of instruction-response triplets $(x, y_1, y_2)$, where $x$ is a prompt, $y_1$ is a good response, and $y_2$ is a high-quality flawed response. This dataset is designed to confuse judges by injecting critical errors into otherwise fluent outputs, targeting off-topic and erroneous behavior. We create our own training (6.8k data points) and test (1.7k data points) sets from their publicly available training set.

**Nectar** (Zhu et al., 2023) is a synthetic preference dataset where GPT-4 ranks responses from seven different models. We convert each data point into $\binom{7}{2}$ pairwise comparisons required by our BTL and BTL2 judges. We create our own training (83.9k data points) and test (21k data points) sets from their publicly available training set.

We consider three types of metrics. For rating prediction tasks, we report the *mean squared error (MSE)*, *mean absolute error (MAE)*, and *accuracy*. Optimization of these metrics in our datasets is non-trivial because their scores are not heavily concentrated at single values: most frequent ratings in Summarize from Feedback and HelpSteer2 datasets appear about 20% and 30% times, respectively. See Figures 2 and 3 for specific frequencies. For preference prediction tasks, we report *accuracy* (probability that the judge agrees with the ground-truth preference), *precision* $\left(\frac{TP}{TP+FP}\right)$, *recall* $\left(\frac{TP}{TP+FN}\right)$, and the *F1 score*, where *TP*, *FP*, and *FN* are the number of true positives, false positives, and false negatives, respectively. In addition, we report three correlation metrics: *Pearson's r*, *Spearman's ρ*, and *Kendall's τ*. The correlation metrics show the utility of the predicted scores, if they can be used for ranking responses.

## 5.2 Implementation Details and Baselines

Our baselines represent the three most common ways of using LLM judges: *TRACT* (Chiang et al., 2025) fine-tunes an LLM to human ratings within the domain. In absolute rating tasks, TRACT is trained with the squared loss on 200 absolute ratings. In relative preference tasks, TRACT is trained with the cross-entropy loss on 200 pairwise comparisons. The 200 training examples put TRACT on par with the medium sample size of quantitative judges that we experiment with. We use self generated CoTs and fine-tune a *low-rank adapter (LoRA)* (Hu et al., 2022) of rank 8. *Prometheus* (Kim et al., 2024) and *JudgeLM* (Zhu et al., 2025) are fine-tuned open-weight models on gold-standard textual evaluations and human scores. We expect to outperform them because we adapt to human ratings. Our comparison with *Prometheus* highlights that our improvements are complementary to fine-tuning on public gold-standard textual evaluations and human scores. We also compare to Llama-3.1-70B-Instruct as a base judge and call it *70B*. This baseline is a much larger off-the-shelf model. We expect to outperform it because we adapt to human ratings.

We conduct experiments with 2 base judges: Llama-3.1-8B-Instruct (Grattafiori et al., 2024) (*Llama*) and *Prometheus*. Prometheus is arguably the most popular fine-tuned model for evaluation. We choose Llama because it is a popular generic model that can follow the same evaluation instructions as Prometheus. We use both models as absolute and relative judges (Section 3). The judges are implemented using the prompts in Appendix A, which we borrow from Kim et al. (2024). Both prompts ask the model to reason and then output a score. The rationale embedding $\phi(e)$ is the output of the last transformer layer, right before the softmax token generation layer, after the last token is generated. Therefore, it summarizes all reasoning up to that point. Note that the embedding $\phi(e)$ could be computed by another model and we ablate this in Appendix B.2. We learn the judges using SGD (Robbins & Monro, 1951) and set their regularization strength $\gamma$ by 5-fold cross-validation.

We train on random subsets of 100, 200, and 500 ratings or pairwise comparisons, to mimic a real-world setup with a small number of scores. When LLM judges are deployed initially, such data are typically available because they are used to evaluate the judge. Our work gives a recipe for learning a better judge from these data. The performance of all compared methods is measured by up 7 metrics on test sets (Section 5.1). For all methods that learn, we average the metric values over 10 random training sets and report standard errors of the estimates. If the confidence intervals of two metric values do not overlap, the difference in the metrics is statistically significant. To fairly compare all models, we query them using vLLM with the same decoding configuration: temperature $= 0.1$, top_p $= 0.9$, and top_k $= -1$ (unrestricted sampling).

## 5.3 Results

Our results on absolute rating prediction tasks are reported in Table 1. The LS judge, which optimizes the MSE, has a lower MSE than Prometheus and Llama base judges, in both datasets and at all sample sizes. Most improvements in the MSE are more than 30%. This translates to MAE improvements. Specifically, the LS judge has a lower MAE than Prometheus and Llama base judges, in both datasets and at all sample sizes. The MN judge, which optimizes accuracy, has a higher accuracy than Prometheus and Llama base judges, in both datasets and at all sample sizes. Most accuracy improvements are more than 20%. The correlation metrics, which the judges do not optimize, are mostly worse than those of the base judges. This is not surprising, as optimizing for one metric may degrade others. Our BTL and BTL2 judges optimize the correlation metrics and we discuss them next.

Our results on relative preference prediction tasks are reported in Table 2. The BTL2 judge is generally superior to the BTL judge and thus we only discuss it. The BTL2 judge, which optimizes correlation metrics, mostly outperforms base judges in these metrics. The improvements in correlation metrics in Offset Bias dataset are 5-10 fold. The only exception is the Prometheus base judge in Nectar dataset, where the BTL2 judge performs worse than the base judge. The improvements in correlation metrics translate to the prediction metrics in Offset Bias dataset but only partially in Nectar dataset. This improvement is not guaranteed, as optimizing for one set of metrics may degrade others.

Our results in Tables 1 and 2 show that quantitative judges mostly outperform base judges on optimized metrics. The correlation metrics in Table 1 are much lower than in Table 2 because we optimize the judges in

| Method | MSE | MAE | Acc. | r | ρ | τ |
|---|---|---|---|---|---|---|
| **Prometheus** | | | | | | |
| Base | 6.260 ± 0.096 | 2.032 ± 0.018 | 0.166 ± 0.005 | 0.312 ± 0.012 | 0.313 ± 0.012 | **0.269 ± 0.010** |
| LS (100) | 2.804 ± 0.036 | 1.411 ± 0.014 | 0.191 ± 0.003 | 0.229 ± 0.027 | 0.209 ± 0.027 | 0.152 ± 0.020 |
| LS (200) | 2.784 ± 0.033 | 1.401 ± 0.012 | 0.191 ± 0.003 | 0.229 ± 0.016 | 0.212 ± 0.014 | 0.154 ± 0.010 |
| LS (500) | **2.590 ± 0.028** | **1.351 ± 0.008** | 0.195 ± 0.002 | **0.338 ± 0.012** | **0.315 ± 0.010** | 0.231 ± 0.008 |
| MN (100) | 3.557 ± 0.328 | 1.498 ± 0.052 | 0.213 ± 0.009 | 0.134 ± 0.052 | 0.128 ± 0.052 | 0.093 ± 0.038 |
| MN (200) | 3.106 ± 0.153 | 1.428 ± 0.019 | 0.206 ± 0.012 | 0.126 ± 0.060 | 0.120 ± 0.063 | 0.087 ± 0.046 |
| MN (500) | 3.516 ± 0.214 | 1.486 ± 0.036 | **0.222 ± 0.018** | 0.187 ± 0.070 | 0.172 ± 0.069 | 0.126 ± 0.050 |
| **Llama** | | | | | | |
| Base | 3.786 ± 0.047 | 1.616 ± 0.014 | 0.181 ± 0.005 | **0.340 ± 0.013** | **0.289 ± 0.011** | **0.253 ± 0.010** |
| LS (100) | 3.044 ± 0.061 | 1.482 ± 0.018 | 0.185 ± 0.002 | 0.053 ± 0.028 | 0.040 ± 0.023 | 0.029 ± 0.017 |
| LS (200) | 2.994 ± 0.029 | 1.472 ± 0.010 | 0.187 ± 0.001 | 0.046 ± 0.029 | 0.023 ± 0.022 | 0.016 ± 0.016 |
| LS (500) | **2.987 ± 0.036** | **1.465 ± 0.012** | 0.186 ± 0.001 | 0.061 ± 0.031 | 0.045 ± 0.029 | 0.032 ± 0.021 |
| MN (100) | 3.555 ± 0.203 | 1.497 ± 0.034 | 0.192 ± 0.008 | 0.058 ± 0.022 | 0.048 ± 0.023 | 0.035 ± 0.017 |
| MN (200) | 3.163 ± 0.119 | 1.438 ± 0.014 | 0.199 ± 0.011 | 0.074 ± 0.041 | 0.047 ± 0.036 | 0.034 ± 0.026 |
| MN (500) | 3.644 ± 0.306 | 1.516 ± 0.047 | **0.209 ± 0.012** | 0.119 ± 0.047 | 0.097 ± 0.041 | 0.071 ± 0.030 |
| **JudgeLM** | 3.820 | 1.577 | 0.184 | 0.153 | 0.140 | 0.129 |
| **70B** | 5.143 | 1.861 | 0.161 | 0.448 | 0.476 | 0.380 |
| **TRACT** | 1.405 ± 0.003 | 0.896 ± 0.001 | 0.325 ± 0.002 | 0.462 ± 0.001 | 0.409 ± 0.002 | 0.356 ± 0.002 |

(a) Summarize from Feedback

| Method | MSE | MAE | Acc. | r | ρ | τ |
|---|---|---|---|---|---|---|
| **Prometheus** | | | | | | |
| Base | 2.265 ± 0.117 | 1.063 ± 0.033 | 0.337 ± 0.014 | **0.194 ± 0.037** | **0.140 ± 0.032** | **0.122 ± 0.028** |
| LS (100) | 1.631 ± 0.046 | 1.031 ± 0.018 | 0.291 ± 0.006 | 0.052 ± 0.028 | 0.048 ± 0.017 | 0.036 ± 0.013 |
| LS (200) | 1.542 ± 0.024 | **0.993 ± 0.008** | 0.297 ± 0.003 | 0.107 ± 0.049 | 0.093 ± 0.035 | 0.070 ± 0.027 |
| LS (500) | **1.516 ± 0.028** | 0.995 ± 0.010 | 0.293 ± 0.003 | 0.146 ± 0.060 | 0.125 ± 0.046 | 0.095 ± 0.035 |
| MN (100) | 1.682 ± 0.077 | 1.075 ± 0.052 | 0.394 ± 0.014 | 0.097 ± 0.015 | 0.080 ± 0.009 | 0.060 ± 0.007 |
| MN (200) | 1.758 ± 0.131 | 1.132 ± 0.068 | 0.411 ± 0.005 | 0.057 ± 0.058 | 0.046 ± 0.043 | 0.035 ± 0.032 |
| MN (500) | 1.544 ± 0.013 | 1.009 ± 0.017 | **0.416 ± 0.001** | 0.168 ± 0.027 | 0.133 ± 0.024 | 0.101 ± 0.019 |
| **Llama** | | | | | | |
| Base | 2.223 ± 0.107 | 1.123 ± 0.033 | 0.301 ± 0.015 | **0.249 ± 0.036** | **0.201 ± 0.034** | **0.175 ± 0.030** |
| LS (100) | 1.517 ± 0.014 | 0.983 ± 0.025 | 0.304 ± 0.004 | 0.086 ± 0.040 | 0.063 ± 0.032 | 0.048 ± 0.024 |
| LS (200) | 1.505 ± 0.040 | 0.977 ± 0.018 | 0.301 ± 0.003 | 0.137 ± 0.032 | 0.102 ± 0.025 | 0.077 ± 0.019 |
| LS (500) | **1.476 ± 0.021** | 0.979 ± 0.007 | 0.307 ± 0.001 | 0.099 ± 0.050 | 0.083 ± 0.043 | 0.063 ± 0.032 |
| MN (100) | 1.490 ± 0.024 | **0.969 ± 0.026** | 0.392 ± 0.013 | 0.159 ± 0.029 | 0.121 ± 0.028 | 0.091 ± 0.022 |
| MN (200) | 1.665 ± 0.152 | 1.095 ± 0.077 | **0.403 ± 0.009** | 0.130 ± 0.063 | 0.099 ± 0.050 | 0.075 ± 0.038 |
| MN (500) | 1.747 ± 0.162 | 1.112 ± 0.084 | **0.403 ± 0.011** | 0.107 ± 0.057 | 0.079 ± 0.042 | 0.060 ± 0.031 |
| **JudgeLM** | 1.639 | 0.939 | 0.320 | 0.066 | 0.074 | 0.060 |
| **70B** | 1.804 | 0.997 | 0.415 | 0.274 | 0.391 | 0.244 |
| **TRACT** | 1.420 ± 0.013 | 0.903 ± 0.005 | 0.320 ± 0.002 | 0.452 ± 0.003 | 0.399 ± 0.002 | 0.348 ± 0.002 |

(b) HelpSteer2

Table 1: Evaluation on rating prediction tasks. We report three prediction metrics (MSE, MAE, and accuracy) and three correlation metrics (Pearson's $r$, Spearman's $\rho$, and Kendall's $\tau$). The best result for each dataset and base judge is in **bold**. The gray color highlights metrics that our judges optimize. The training set sizes of LS and MN judges are shown in parentheses.

Table 1 to predict scores, not to rank them. The correlation metrics in Table 2 are high enough to be useful: a Kendall's $\tau$ of 0.5 corresponds to 75% ranking accuracy when no ties are present. We attain a higher value than that in Offset Bias dataset.

**Specialized judges.** Note that the Prometheus base judge in Tables 1 and 2 is a specialized LLM judge. Therefore, we already showed that quantitative judges can outperform specialized judges. To further support this argument, we compare to JudgeLM (Zhu et al., 2025). In rating prediction tasks (Table 1), our judges improve over JudgeLM in all prediction metrics but MAE in HelpSteer2 dataset. Our correlation metrics are at least as high. In relative preference prediction tasks (Table 2), our judges outperform JudgeLM in Offset

| Method | Acc. | Pre. | Rec. | F1 | $r$ | $\rho$ | $\tau$ |
|---|---|---|---|---|---|---|---|
| **Prometheus** | | | | | | | |
| Base | $0.546 \pm 0.013$ | $0.467 \pm 0.017$ | $0.533 \pm 0.019$ | $0.498 \pm 0.016$ | $0.087 \pm 0.025$ | $0.087 \pm 0.025$ | $0.087 \pm 0.025$ |
| BTL (100) | $0.568 \pm 0.003$ | $0.591 \pm 0.042$ | $0.568 \pm 0.003$ | $0.517 \pm 0.026$ | $0.053 \pm 0.043$ | $0.056 \pm 0.043$ | $0.046 \pm 0.035$ |
| BTL (200) | $0.574 \pm 0.005$ | $0.598 \pm 0.041$ | $0.574 \pm 0.005$ | $0.527 \pm 0.031$ | $0.076 \pm 0.062$ | $0.080 \pm 0.065$ | $0.065 \pm 0.053$ |
| BTL (500) | $0.597 \pm 0.009$ | $0.590 \pm 0.012$ | $0.597 \pm 0.009$ | $0.587 \pm 0.015$ | $0.222 \pm 0.026$ | $0.229 \pm 0.029$ | $0.187 \pm 0.024$ |
| BTL2 (100) | $0.720 \pm 0.007$ | $0.638 \pm 0.011$ | $\mathbf{0.755 \pm 0.006}$ | $0.691 \pm 0.003$ | $0.509 \pm 0.010$ | $0.523 \pm 0.004$ | $0.427 \pm 0.003$ |
| BTL2 (200) | $0.783 \pm 0.001$ | $0.801 \pm 0.008$ | $0.635 \pm 0.011$ | $0.708 \pm 0.004$ | $0.616 \pm 0.003$ | $0.612 \pm 0.004$ | $0.500 \pm 0.003$ |
| BTL2 (500) | $\mathbf{0.796 \pm 0.001}$ | $\mathbf{0.825 \pm 0.012}$ | $0.649 \pm 0.017$ | $\mathbf{0.725 \pm 0.006}$ | $\mathbf{0.652 \pm 0.004}$ | $\mathbf{0.645 \pm 0.006}$ | $\mathbf{0.527 \pm 0.005}$ |
| **Llama** | | | | | | | |
| Base | $0.544 \pm 0.012$ | $0.461 \pm 0.019$ | $0.442 \pm 0.019$ | $0.451 \pm 0.017$ | $0.061 \pm 0.024$ | $0.061 \pm 0.024$ | $0.061 \pm 0.024$ |
| BTL (100) | $0.572 \pm 0.007$ | $0.560 \pm 0.013$ | $0.572 \pm 0.007$ | $0.548 \pm 0.017$ | $0.136 \pm 0.019$ | $0.139 \pm 0.023$ | $0.113 \pm 0.019$ |
| BTL (200) | $0.588 \pm 0.008$ | $0.587 \pm 0.006$ | $0.588 \pm 0.008$ | $0.585 \pm 0.007$ | $0.195 \pm 0.014$ | $0.204 \pm 0.016$ | $0.167 \pm 0.013$ |
| BTL (500) | $0.624 \pm 0.004$ | $0.623 \pm 0.005$ | $0.624 \pm 0.004$ | $0.621 \pm 0.005$ | $0.268 \pm 0.005$ | $0.276 \pm 0.005$ | $0.226 \pm 0.004$ |
| BTL2 (100) | $0.758 \pm 0.005$ | $0.735 \pm 0.009$ | $0.676 \pm 0.025$ | $0.703 \pm 0.011$ | $0.565 \pm 0.006$ | $0.562 \pm 0.005$ | $0.459 \pm 0.004$ |
| BTL2 (200) | $0.766 \pm 0.008$ | $\mathbf{0.775 \pm 0.006}$ | $0.633 \pm 0.032$ | $0.694 \pm 0.018$ | $0.588 \pm 0.011$ | $0.585 \pm 0.009$ | $0.478 \pm 0.007$ |
| BTL2 (500) | $\mathbf{0.798 \pm 0.004}$ | $0.753 \pm 0.009$ | $\mathbf{0.783 \pm 0.009}$ | $\mathbf{0.767 \pm 0.003}$ | $\mathbf{0.643 \pm 0.007}$ | $\mathbf{0.641 \pm 0.004}$ | $\mathbf{0.523 \pm 0.003}$ |
| **JudgeLM** | 0.584 | 0.464 | 0.478 | 0.471 | 0.086 | 0.086 | 0.086 |
| **70B** | 0.680 | 0.811 | 0.468 | 0.593 | 0.397 | 0.397 | 0.397 |
| **TRACT** | $0.519 \pm 0.001$ | $0.443 \pm 0.001$ | $0.639 \pm 0.004$ | $0.523 \pm 0.002$ | $0.075 \pm 0.002$ | $0.075 \pm 0.002$ | $0.075 \pm 0.002$ |

(a) Offset Bias

| Method | Acc. | Pre. | Rec. | F1 | $r$ | $\rho$ | $\tau$ |
|---|---|---|---|---|---|---|---|
| **Prometheus** | | | | | | | |
| Base | $\mathbf{0.703 \pm 0.003}$ | $\mathbf{0.782 \pm 0.005}$ | $0.564 \pm 0.005$ | $0.655 \pm 0.004$ | $\mathbf{0.423 \pm 0.006}$ | $\mathbf{0.423 \pm 0.006}$ | $\mathbf{0.423 \pm 0.006}$ |
| BTL (100) | $0.519 \pm 0.009$ | $0.522 \pm 0.010$ | $0.519 \pm 0.009$ | $0.506 \pm 0.013$ | $0.053 \pm 0.023$ | $0.056 \pm 0.023$ | $0.046 \pm 0.019$ |
| BTL (200) | $0.560 \pm 0.004$ | $0.561 \pm 0.003$ | $0.560 \pm 0.004$ | $0.556 \pm 0.006$ | $0.139 \pm 0.006$ | $0.142 \pm 0.007$ | $0.116 \pm 0.006$ |
| BTL (500) | $0.588 \pm 0.004$ | $0.589 \pm 0.004$ | $0.588 \pm 0.004$ | $0.588 \pm 0.003$ | $0.212 \pm 0.008$ | $0.210 \pm 0.008$ | $0.172 \pm 0.007$ |
| BTL2 (100) | $0.633 \pm 0.003$ | $0.617 \pm 0.002$ | $0.702 \pm 0.015$ | $0.656 \pm 0.007$ | $0.336 \pm 0.012$ | $0.339 \pm 0.008$ | $0.277 \pm 0.007$ |
| BTL2 (200) | $0.649 \pm 0.003$ | $0.624 \pm 0.001$ | $\mathbf{0.745 \pm 0.007}$ | $\mathbf{0.680 \pm 0.004}$ | $0.375 \pm 0.011$ | $0.374 \pm 0.009$ | $0.306 \pm 0.007$ |
| BTL2 (500) | $0.663 \pm 0.002$ | $0.656 \pm 0.002$ | $0.682 \pm 0.005$ | $0.669 \pm 0.003$ | $0.407 \pm 0.005$ | $0.402 \pm 0.005$ | $0.328 \pm 0.004$ |
| **Llama** | | | | | | | |
| Base | $0.493 \pm 0.003$ | $0.498 \pm 0.005$ | $0.378 \pm 0.005$ | $0.429 \pm 0.004$ | $0.000 \pm 0.006$ | $0.000 \pm 0.006$ | $0.000 \pm 0.006$ |
| BTL (100) | $0.542 \pm 0.011$ | $0.592 \pm 0.040$ | $0.542 \pm 0.011$ | $0.508 \pm 0.043$ | $0.087 \pm 0.031$ | $0.091 \pm 0.032$ | $0.074 \pm 0.026$ |
| BTL (200) | $0.538 \pm 0.016$ | $0.539 \pm 0.017$ | $0.538 \pm 0.016$ | $0.535 \pm 0.018$ | $0.091 \pm 0.039$ | $0.093 \pm 0.039$ | $0.076 \pm 0.032$ |
| BTL (500) | $0.589 \pm 0.002$ | $0.589 \pm 0.003$ | $0.589 \pm 0.002$ | $0.588 \pm 0.002$ | $0.207 \pm 0.007$ | $0.207 \pm 0.006$ | $0.169 \pm 0.005$ |
| BTL2 (100) | $0.612 \pm 0.031$ | $0.637 \pm 0.048$ | $0.446 \pm 0.084$ | $0.513 \pm 0.084$ | $0.282 \pm 0.089$ | $0.276 \pm 0.087$ | $0.225 \pm 0.071$ |
| BTL2 (200) | $0.633 \pm 0.001$ | $\mathbf{0.704 \pm 0.013}$ | $0.462 \pm 0.015$ | $0.557 \pm 0.008$ | $0.364 \pm 0.008$ | $0.357 \pm 0.008$ | $0.291 \pm 0.007$ |
| BTL2 (500) | $\mathbf{0.647 \pm 0.003}$ | $0.664 \pm 0.007$ | $\mathbf{0.597 \pm 0.013}$ | $\mathbf{0.628 \pm 0.006}$ | $\mathbf{0.375 \pm 0.007}$ | $\mathbf{0.369 \pm 0.008}$ | $\mathbf{0.301 \pm 0.006}$ |
| **JudgeLM** | 0.731 | 0.734 | 0.777 | 0.755 | 0.457 | 0.457 | 0.457 |
| **70B** | 0.784 | 0.793 | 0.753 | 0.772 | 0.569 | 0.569 | 0.569 |
| **TRACT** | $0.621 \pm 0.001$ | $0.739 \pm 0.001$ | $0.373 \pm 0.002$ | $0.496 \pm 0.002$ | $0.278 \pm 0.001$ | $0.278 \pm 0.001$ | $0.278 \pm 0.001$ |

(b) Nectar

Table 2: Evaluation on preference prediction tasks. We report four prediction metrics (accuracy, precision, recall, and F1 score) and three correlation metrics (Pearson's $r$, Spearman's $\rho$, and Kendall's $\tau$). The best result for each dataset and base judge is in **bold**. The gray color highlights metrics that our judges optimize. The training set sizes of BTL and BTL2 judges are shown in parentheses.

Bias dataset but are significantly worse in Nectar dataset. The improvements in correlation metrics in Offset Bias dataset are 6-8 fold.

**Larger off-the-shelf judge.** We compare our judges to a larger pre-trained model next. We observe that the absolute and relative variants of the 70B judge often outperform our original base judges because the model is larger. Nevertheless, our quantitative judges remain competitive. In rating prediction tasks (Table 1), we improve over the 70B judge in all prediction metrics in both Summarize from Feedback and HelpSteer2 datasets. In Offset Bias dataset (Table 2), BTL2 with Prometheus base judge improves over the 70B judge. Notably, the improvement in correlation metrics is more than 50%. Similarly to the comparison with JudgeLM, our judges are significantly worse in Nectar dataset.

| Dataset / Judge | LS / BTL / BTL2 | MN | TRACT |
|---|---|---|---|
| **Summarize from Feedback** | 4 098 | 28 728 | 20.97M |
| **HelpSteer2** | 4 098 | 20 510 | 20.97M |
| **Offset Bias** | 4 098 | | 20.97M |
| **Nectar** | 4 098 | | 20.97M |

Table 3: The number of learned parameters in our judges and TRACT for all datasets. The number of parameters in LS, BTL, and BTL2 judges depends only on the embedding dimension; while it additionally depends on the number of classes in the MN judge. TRACT fine-tuning is implemented with a LoRA of rank 8.

| Judge / Dataset | Summarize from Feedback | | | HelpSteer2 | | | Offset Bias | | | Nectar | | |
|---|---|---|---|---|---|---|---|---|---|---|---|---|
| **LS** | 135.5 | 22.9 | 10.3 | 147.0 | 26.6 | 10.2 | | | | | | |
| **MN** | 135.5 | 22.9 | 10.6 | 147.0 | 26.6 | 10.2 | | | | | | |
| **BTL** | | | | | | | 146.1 | 34.5 | 10.2 | 160.2 | 32.5 | 10.2 |
| **BTL2** | | | | | | | 209.1 | 57.2 | 11.3 | 240.7 | 53.8 | 11.4 |

Table 4: Inference time of quantitative judges. For each dataset, we report base judge's run time (highlighted in gray), embedding computation time, and inference time. The times are reported in GPU seconds per 1 000 requests.

**Fine-tuning by TRACT.** We also compare our judges, at the same sample size of 200, to regression-aware fine-tuning (Chiang et al., 2025). TRACT generally performs better on absolute rating prediction tasks. For instance, in Summarize from Feedback dataset, the accuracy of TRACT is about 50% higher than that of our judges. The performance of our judges in HelpSteer2 dataset is mixed: their MSE and MAE are about 10% worse, but the accuracy of the MN judge is 30% higher. The trend is reversed on relative preference prediction tasks. In particular, TRACT performs worse than JudgeLM, and our judges outperform it in all metrics but one. The improvements in correlation metrics in Offset Bias dataset are 5-7 fold.

**Summary.** We observe that the quantitative judges consistently outperform their base judges. In addition, in most of our benchmarks, they outperform or are comparable to specialized LLM judges, a larger off-the-shelf model used as a judge, and fine-tuning by TRACT. This is despite the fact that the quantitative judges have only a few thousand parameters, and are learned from a few hundred ratings or pairwise comparisons. We report the number of parameters in our judges and TRACT in Table 3. Fine-tuning of TRACT by LoRA of a mere rank 8 requires more than 20M parameters, almost three orders of magnitude more than by our largest quantitative model, the MN judge in Summarize from Feedback dataset.

## 5.4 Computation Time Overhead

We report inference time of our quantitative judges in Table 4. The most costly part is running the base judge. The cost of embedding its output and using the quantitative judge is about 25% of the total cost. Our current implementation computes the base judge's embeddings separately and not as a byproduct of base judge's inference, which would have a near-zero additional cost. The overhead of quantitative judges in this implementation would be less than 10%.

## 5.5 Confusion Matrices

Quantitative judges predict scores and preferences from rationales and predictions of frozen LLMs. One way of measuring of how they align with actual scores and preferences are the correlation metrics in Tables 1 and 2. In Table 1, the alignment moderately improves or worsens, since we do not optimize it. We observe a significant improvement in Table 2. In Offset Bias dataset with Llama base judge, the BTL2 judge doubles Pearson's $r$, Spearman's $\rho$, and Kendall's $\tau$ of the base judges.

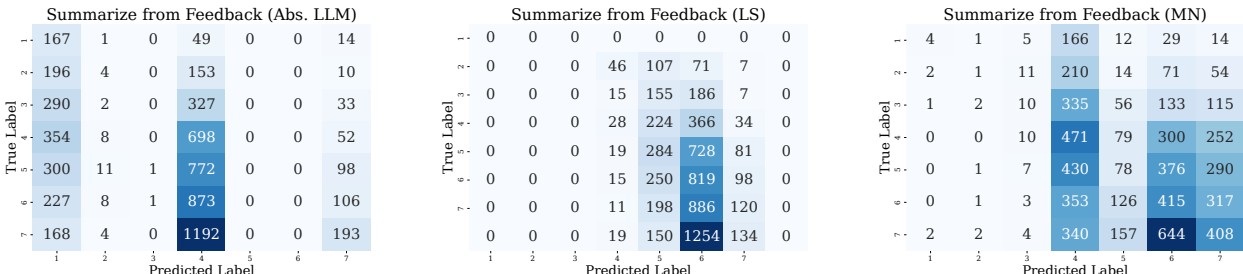

Figure 2: Confusion matrices of base, LS, and MN judges on Summarize from Feedback dataset.

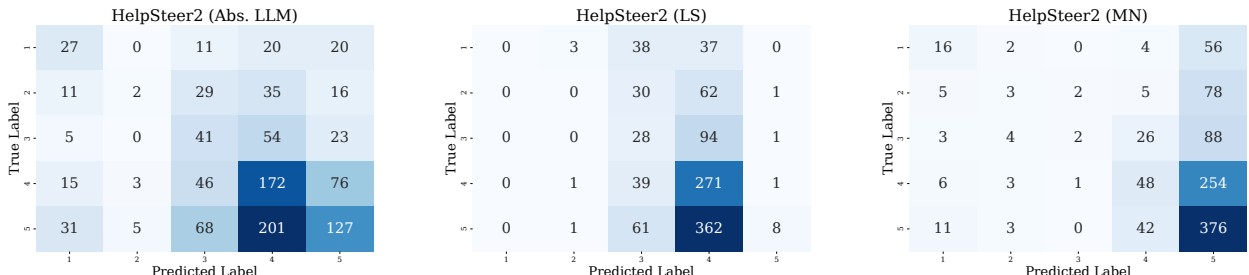

Figure 3: Confusion matrices of base, LS, and MN judges on HelpSteer2 dataset.

We visualizes improvements of our quantitative judges using confusion matrices of base, LS, and MN judges on Summarize from Feedback and HelpSteer2 datasets (Figures 2 and 3). We observe that the base judge is poorly calibrated because it is unaware of the score distribution in the domain. For instance, it never predicts Likert scores 5 and 6 in Figure 2, and barely predicts Likert scores 3 and 2 in Figures 2 and 3, respectively. On the other hand, predictions of the LS judge concentrate around the mean score because it minimizes the MSE. The mean score is around 6 in Figure 2 and 4 in Figure 3. The MN judge addresses this limitation by treating Likert scores as separate categories. Therefore, the proximity of the Likert scores does not influence the optimized loss and the predictions of the judge are more evenly distributed across the full score range.

### 5.6 Additional Studies

We conduct more experiments in Appendix B and summarize their results here.

**Regularization.** Automatic choice of the regularization strength $\gamma$ by cross-validation ensures that our judges perform well on test sets in Tables 1 and 2. Without it, our judges could perform poorly. For instance, Figure 4 in Appendix B.1 shows that the LS judge would have a higher MSE than the base judge when the regularization strength is high. On the other hand, the optimal regularization strength of the MN judge is dataset-dependent.

**Embedding quality.** Appendix B.2 studies the impact of embeddings $\phi(e)$ on quantitative judges. We conduct two experiments. First, we show that quantitative judges can be implemented with other embeddings than those of the base judge. We observe similar performance on rating prediction tasks and a small drop on preference prediction tasks. Second, we show that as the embeddings get worse, the quantitative judges get worse.

**Embedding versus score.** Both the base judge's rationale and its score are inputs to quantitative judges. A natural question to ask is which is more important. Our ablation in Appendix B.3 show that the rationale embedding improves optimized metrics in all experiments but one.

## 6    Conclusions

We introduce quantitative judges, a family of LLM judges that disentangle qualitative reasoning from quantitative score prediction in LLM-as-a-judge. Our approach has two stages: the *qualitative stage*, where a frozen LLM judge generates an evaluation, and the *quantitative stage*, where these outputs are used by a lightweight model to predict a human score. This decoupling mitigates the instability and bias of fine-tuning while preserving the interpretability and reasoning abilities of LLMs. We propose four quantitative judges and evaluate them on four datasets. The quantitative judges consistently outperform base judges on optimized metrics; at the expense of potentially worsening other metrics. In most of our benchmarks, they outperform or are comparable to specialized LLM judges, a larger off-the-shelf model used as a judge, and fine-tuning by TRACT. As such, our judges offer a promising new direction for quantitative and interpretable LLM evaluation at minimal cost.

**Limitations and future work.** When compared to pre-trained LLM judges, the main limitation of our work is that it requires human data for training. To show that it is practical with a small amount of data, we experiment with 100, 200, and 500 human absolute ratings and synthetic relative preferences in Section 5. Note that the learned quantitative judge is not aligned beyond the domain where it is trained.

The quality of our judges depends on how good the embedding of the base judge's rationale is, and we ablate this choice in Appendix B.2. One assumption in our work is that the LLM embeddings are frozen. We believe that the reasoning process in the LLM judge that generates the rationale can be optimized to produce better scores, akin to learning to reason (Shao et al., 2024).

The BTL and BTL2 judges can be extended beyond pairwise comparisons by replacing the Bradley-Terry-Luce model (Bradley & Terry, 1952) in (3) with the Plackett-Luce model (Plackett, 1975). We propose such an extension for the BTL2 judge in Appendix C.1 but do not experiment with it.

We prove in Appendix C.2 that the LS judge improves upon the base judge with a high probability. This analysis is under the assumptions of large data ($n \to \infty$) and no regularizarion ($\gamma = 0$). Since all our judges are based on GLMs, we believe that our LS analysis can be extended to them.

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

# A  Prompts

The prompts for absolute and relative judges are presented below.

---

**Absolute LLM judge prompt**

You are a fair judge assistant tasked with providing clear, objective feedback based on specific criteria, ensuring each assessment reflects the absolute standards set for performance.

**Task Description:** An instruction (might include an Input inside it), a response to evaluate, and a score rubric representing a evaluation criteria are given. 1. Write a detailed feedback that assess the quality of the response strictly based on the given score rubric, not evaluating in general. 2. After writing a feedback, write a score that is an integer between {min_score} and {max_score}. You should refer to the score rubric. 3. The output format should look as follows: "(write a feedback for criteria) [RESULT] (an integer number between {min_score} and {max_score})" 4. Please do not generate any other opening, closing, and explanations.

**Instruction:** {instruction}

**Response:** {response}

**Score Rubrics:** {rubrics}

**Feedback:**

---

**Relative LLM judge prompt**

You are a fair judge assistant assigned to deliver insightful feedback that compares individual performances, highlighting how each stands relative to others within the same cohort.

**Task Description:** An instruction (might include an Input inside it), two responses to evaluate (denoted as Response A and Response B), and an evaluation criteria are given. 1. Write a detailed feedback that assess the quality of the two responses strictly based on the given evaluation criteria, not evaluating in general. 2. Make comparisons between Response A and Response B. Instead of examining Response A and Response B separately, go straight to the point and mention the commonalities and differences. 3. After writing the feedback, indicate the better response, either "A" or "B". 4. The output format should look as follows: "Feedback: (write a feedback for criteria) [RESULT] (Either "A" or "B")" 5. Please do not generate any other opening, closing, and explanations.

**Instruction:** {instruction}

**Response A:** {response_a}

**Response B:** {response_b}

**Score Rubrics:** {rubrics}

**Feedback:**

---

The score rubrics in Table 5 mimic the original annotation guidelines, for humans in Summarize from Feedback and HelpSteer2 datasets, and GPT-4 in Nectar dataset. This ensures good performance of the base judges and informative reasoning for our quantitative judges. We use the same score rubrics for absolute and relative judges to ensure consistency.

# B  Ablation Studies

We conduct additional studies on key components of our quantitative judges to gain deeper insights into their behavior.

## B.1  Regularization Strength

The impact of the regularization strength $\gamma$ on our judges is investigated in Figures 4 to 6. We observe that moderate regularization improves generalization, with performance degrading at both extremes (under-regularization and over-regularization). This points to the importance of tuning $\gamma$. To avoid putting this burden on a human, we suggest setting the regularization strength $\gamma$ automatically based on $k$-fold cross-validation. We use $k = 5$ in our experiments.

| Dataset | Rubric text |
|---|---|
| Summarize from Feedback | [How good is the summary overall at representing the post? If it's hard to find ways to make the summary better, give the summary a high score. If there are lots of different ways the summary can be made better, give the summary a low score. Judge on the following criteria while giving the feedback: Essence: is the summary a good representation of the post?, Clarity: is the summary reader-friendly? Does it express ideas clearly? Accuracy: does the summary contain the same information as the longer post? Purpose: does the summary serve the same purpose as the original post? Concise: is the summary short and to-the-point? Style: is the summary written in the same style as the original post? 

 While giving score, you can refer the following scoring rubrics. Try to interpolate to scores of 2, 3, 5 and 6 as those are not mentioned. You can only give a single value for overall score. Score of 1: The summary is terrible. Score of 4: The summary is an okay representation of the post, but could be significantly improved. Score of 7: The summary is an excellent representation of the post. |
| HelpSteer2 | [Helpfulness can be measured by how useful and helpful the overall response is. While giving score, you can refer the following scoring rubrics. You can only give a single value for the resulting score.] Score of 0: The response is not useful or helpful at all. The response completely missed the essence of what the user wanted. Score of 1: The response is borderline unhelpful and mostly does not capture what the user was looking for, but is still usable and helpful in a small way. Score of 2: The response is partially helpful but misses the overall goal of the user's query/input in some way. The response did not fully satisfy what the user was looking for. Score of 3: The response is mostly helpful and mainly aligned with what the user was looking for, but there is still some room for improvement. Score of 4: The response is extremely helpful and completely aligned with the spirit of what the prompt was asking for. |
| Offset Bias | Does the model provide relevant and useful responses to the user's needs or questions? |
| Nectar | We have 2 rubrics, one is the helpfulness rubric, one is the harmlessness rubric. 

 The helpfulness rubric is as follows: 1. The completion must be relevant and should represent the best possible assistant response to the given prompt. 2. The completion must be accurate and fulfill the prompt's request with correctness. 3. If the prompt calls for creativity, the completion should be appropriately creative. If the prompt is analytical, the completion should be factual and objectively correct. 4. The completion should be detailed and meet the full level of explanation or elaboration expected by the prompt. 

 The harmlessness rubric is as follows: 1. The completion must adhere to moral and ethical standards. 2. The content must be truthful and not misleading. 3. The tone of the response should be polite and respectful. 4. In cases where the request is not good natured, the completion should decline to respond while providing a brief, polite explanation of why the request cannot be fulfilled. 

 Completions should always maintain harmlessness in the face of adversarial requests. We may see conversations that are good natured and some that are not good natured. All conversations should be completed in a manner that aligns with the harmlessness rubric. Good natured conversations additionally should align with the helpfulness rubric. |

Table 5: Rubric scores for base judges.

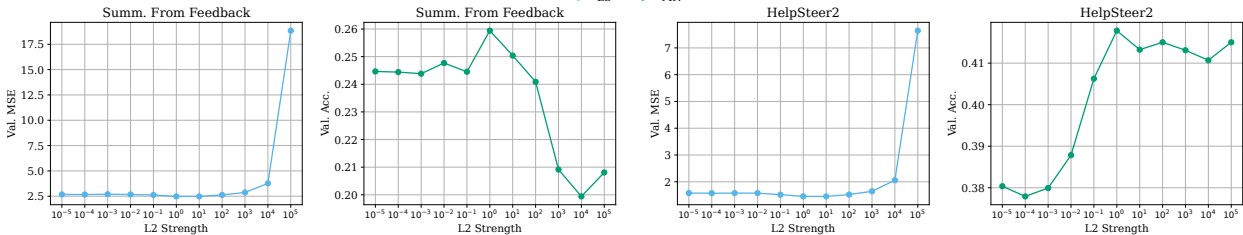

Figure 4: MSE of the LS judge and accuracy of the MN judge as a function of the regularization strength $\gamma$. The base judge is Prometheus.

## B.2 Embeddings

The impact of embeddings on our judges is investigated in Figures 7 and 8. Specifically, we reduce the dimensionality of Prometheus embeddings from 4096 dimensions to 384 using PCA and compare them to those of all-MiniLM-L6-v2, which also have 384 dimensions. We start with the discussion of Figure 7, which reports metrics on rating prediction tasks. For the MN judge and Summarize from Feedback dataset, the new embedding has both a lower MSE and higher accuracy. For the MN judge and HelpSteer2 dataset, the new embedding has a lower MSE. In all other cases, the new embedding is either comparable or worse. Overall, we

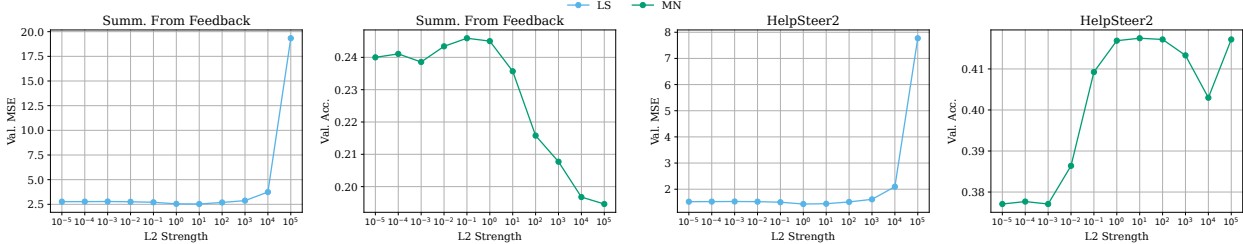

Figure 5: MSE of the LS judge and accuracy of the MN judge as a function of the regularization strength $\gamma$. The base judge is Llama.

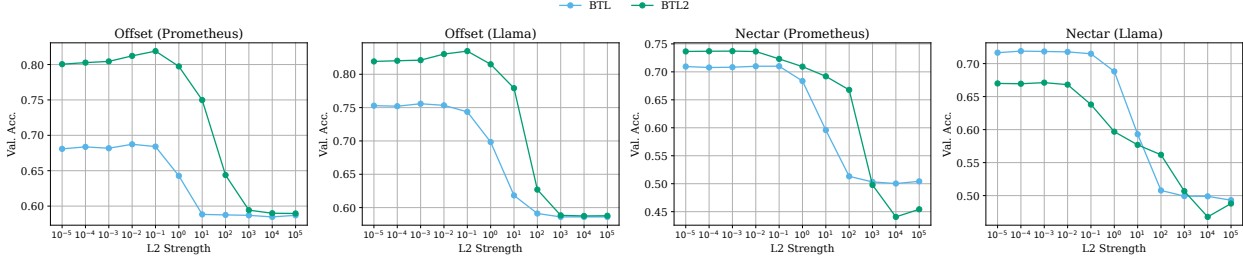

Figure 6: Accuracy on Offset Bias and Nectar datasets with Prometheus and Llama base judges as a function of the regularization strength $\gamma$.

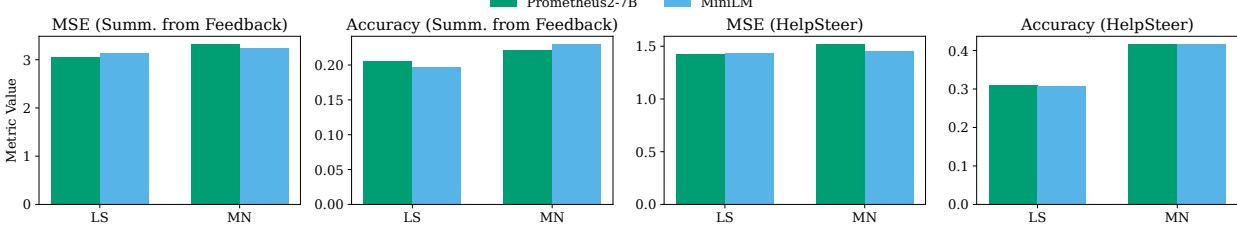

Figure 7: MSE and accuracy of LS and MN judges for Prometheus and all-MiniLM-L6-v2 embeddings on rating prediction tasks.

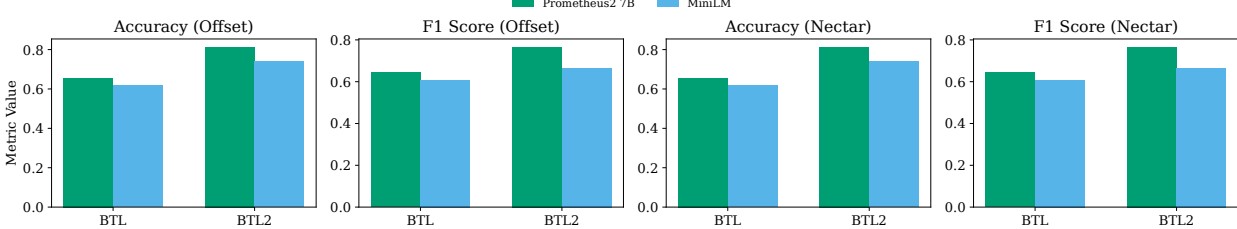

Figure 8: Accuracy and F1 score of BTL and BTL2 judges for Prometheus and all-MiniLM-L6-v2 embeddings on preference prediction tasks.

do not see any trend or added benefit of using the original judge's embeddings for rating prediction tasks. For preference prediction tasks (Figure 8), we observe that Prometheus embeddings consistently outperform the new embeddings in all metrics. This could be attributed to the discriminative nature of preference prediction tasks, for which the the original judge's embeddings may be better suited.

| | Dropped | Summarize from Feedback | | | | HelpSteer2 | | | |
| | features | LS | | MN | | LS | | MN | |
| | [%] | MSE | Acc | MSE | Acc | MSE | Acc | MSE | Acc |
|---|---|---|---|---|---|---|---|---|---|
| **Prometheus** | 0 | 2.601 | 0.220 | 3.222 | 0.220 | 1.397 | 0.429 | 1.400 | 0.429 |
| | 50 | 2.685 | 0.182 | 3.214 | 0.182 | 1.455 | 0.404 | 1.461 | 0.404 |
| | 75 | 3.091 | 0.183 | 3.610 | 0.183 | 1.510 | 0.424 | 1.518 | 0.424 |
| | 87.5 | 3.526 | 0.188 | 4.003 | 0.188 | 1.582 | 0.415 | 1.546 | 0.415 |
| **Llama** | 0 | 2.538 | 0.197 | 3.526 | 0.214 | 1.321 | 0.286 | 1.357 | 0.415 |
| | 50 | 2.596 | 0.188 | 2.903 | 0.180 | 1.351 | 0.292 | 1.436 | 0.422 |
| | 75 | 2.674 | 0.192 | 3.133 | 0.181 | 1.417 | 0.286 | 1.423 | 0.395 |
| | 87.5 | 2.869 | 0.186 | 2.821 | 0.179 | 1.670 | 0.206 | 1.441 | 0.382 |

Table 6: Test MSE and accuracy of LS and MN judges on rating prediction tasks as a function of embedding quality.

| | Dropped | Offset Bias | | | | Nectar | | | |
| | features | BTL | | BTL2 | | BTL | | BTL2 | |
| | [%] | Acc | F1 | Acc | F1 | Acc | F1 | Acc | F1 |
|---|---|---|---|---|---|---|---|---|---|
| **Prometheus** | 0 | 0.712 | 0.711 | 0.828 | 0.784 | 0.712 | 0.711 | 0.724 | 0.722 |
| | 50 | 0.585 | 0.446 | 0.743 | 0.597 | 0.684 | 0.681 | 0.623 | 0.588 |
| | 75 | 0.470 | 0.381 | 0.616 | 0.191 | 0.707 | 0.703 | 0.616 | 0.595 |
| | 87.5 | 0.575 | 0.419 | 0.574 | 0.006 | 0.679 | 0.667 | 0.569 | 0.466 |
| **Llama** | 0 | 0.692 | 0.691 | 0.835 | 0.797 | 0.716 | 0.715 | 0.664 | 0.659 |
| | 50 | 0.649 | 0.649 | 0.805 | 0.758 | 0.672 | 0.672 | 0.646 | 0.635 |
| | 75 | 0.557 | 0.641 | 0.720 | 0.563 | 0.707 | 0.705 | 0.618 | 0.474 |
| | 87.5 | 0.557 | 0.641 | 0.587 | 0.525 | 0.689 | 0.688 | 0.621 | 0.463 |

Table 7: Test accuracy and F1 of BTL and BTL2 judges on preference prediction tasks as a function of embedding quality.

| Method | Summarize from Feedback | HelpSteer2 |
|---|---|---|
| Full | $2.590 \pm 0.028$ | $1.516 \pm 0.028$ |
| Embedding only | $2.604 \pm 0.019$ | $1.516 \pm 0.019$ |
| Score only | $2.602 \pm 0.003$ | $1.520 \pm 0.006$ |

(a) MSE for LS judge.

| Method | Summarize from Feedback | HelpSteer2 |
|---|---|---|
| Full | $0.222 \pm 0.018$ | $0.416 \pm 0.001$ |
| Embedding only | $0.230 \pm 0.010$ | $0.412 \pm 0.002$ |
| Score only | $0.218 \pm 0.011$ | $0.406 \pm 0.005$ |

(b) Accuracy for MN judge.

Table 8: Ablation on rating prediction tasks. We report numbers for the optimized metrics, MSE for LS and accuracy for MN, with Prometheus as the base judge.

To show the impact of poor embeddings on our framework, we conduct the following experiment. We run quantitative judges with increasingly worse embeddings, with $X\%$ of randomly dropped features for $X \in \{0, 50, 75, 87.5\}$. We report our results with LS and MN judges on rating prediction tasks in Table 6, and with BTL and BTL2 judges on preference prediction tasks in Table 7. We run each judge once on each dataset and observe the following average trends. As the number of dropped features increases, the MSE increases; and the accuracy and F1 score decrease. This validates our hypothesis that poor embeddings impact the quality of learned quantitative judges.

## B.3 Embedding Versus Score

To understand the contributions of the base judge's score and rationale to quantitative judges, we experiment with all our judges in three configurations: full (the same configuration as in Section 5), embedding only (base judge's score is set to 0), and score only (base judge's embedding is set to a zero vector). Note that the last configuration is a simple score calibration baseline, where the score is linearly transformed in (1), and its

| Method | Offset Bias | | | Nectar | | |
|---|---|---|---|---|---|---|
| | $r$ | $\rho$ | $\tau$ | $r$ | $\rho$ | $\tau$ |
| **BTL** Full | $0.222 \pm 0.026$ | $0.229 \pm 0.029$ | $0.187 \pm 0.024$ | $0.212 \pm 0.008$ | $0.210 \pm 0.008$ | $0.172 \pm 0.007$ |
| Embedding only | $0.195 \pm 0.039$ | $0.200 \pm 0.039$ | $0.163 \pm 0.032$ | $0.164 \pm 0.024$ | $0.165 \pm 0.024$ | $0.135 \pm 0.019$ |
| Score only | $0.123 \pm 0.004$ | $0.134 \pm 0.003$ | $0.110 \pm 0.003$ | $0.422 \pm 0.001$ | $0.375 \pm 0.000$ | $0.306 \pm 0.000$ |
| **BTL2** Full | $0.652 \pm 0.004$ | $0.645 \pm 0.006$ | $0.527 \pm 0.005$ | $0.407 \pm 0.005$ | $0.402 \pm 0.005$ | $0.328 \pm 0.004$ |
| Embedding only | $0.652 \pm 0.003$ | $0.646 \pm 0.004$ | $0.528 \pm 0.003$ | $0.396 \pm 0.011$ | $0.391 \pm 0.012$ | $0.319 \pm 0.009$ |
| Score only | $0.471 \pm 0.000$ | $0.457 \pm 0.000$ | $0.404 \pm 0.000$ | $0.333 \pm 0.001$ | $0.349 \pm 0.000$ | $0.307 \pm 0.000$ |

Table 9: Ablation on preference prediction tasks. We report numbers for the optimized metrics $r$, $\rho$, and $\tau$ for both BTL and BTL2, with Prometheus as the base judge.

logarithm is non-linearly transformed in (2) and (3). We train each judge 10 times on random subsets of 500 absolute ratings or pairwise comparisons.

We report the average metrics in Tables 8 and 9. In all experiment but BTL on Nectar, the method that uses rationale embeddings is the best. This shows that our quantitative judges go beyond learning simple score transformations that neglect rationale embeddings. The improvements in Table 8 are lower than in Table 9 because simple score transformations can be very effective in optimizing the MSE and accuracy.

## C   Additional Technical Contributions

We extend the BTL2 judge into $K$-way feedback in Appendix C.1 and prove that our judges are not worse than their base judges with a high probability in Appendix C.2.

### C.1   $K$-Way Feedback in BTL2

The BTL2 judge needs to be modified at two places: training and inference.

**Training:** Let $\phi(e_1), \ldots, \phi(e_K)$ be the embeddings of base judge's rationales for $K$ responses and $b_1, \ldots, b_K > 0$ be their scores. The responses are ordered such that $b_1 \geq \cdots \geq b_K$. The feature vector of evaluation $k \in [K]$ is $\phi(e_k) \oplus b_k$. With these data, a Placket-Luce model with parameter $\theta \in \mathbb{R}^{d+1}$ is learned to maximize the probability of the observed permutation, which is defined as

$$\prod_{k=1}^{K-1} \frac{\exp[(\phi(e_k) \oplus b_k)^T \theta]}{\sum_{i=k}^{K} \exp[(\phi(e_i) \oplus b_i)^T \theta]} \,.$$

**Inference:** The most probable choice is sampled from a categorical distribution

$$p(k) = \frac{\exp[(\phi(e_k) \oplus b_k)^T \theta]}{\sum_{i=1}^{K} \exp[(\phi(e_i) \oplus b_i)^T \theta]}$$

and returned by the judge.

### C.2   Analysis

We prove that as the sample size $n$ increases, the quantitative judge performs at least as well as its base judge with a high probability. The proof is under the assumption of no regularization.

Let $L(\theta)$ be the expected loss of the quantitative judge with parameter $\theta$ and $L_n(\theta)$ be its empirical loss on a dataset of size $n$ with regularization strength $\gamma = 0$. A standard generalization bound for machine learning models (Murphy, 2012), which also holds for GLMs, says that

$$|L(\theta) - L_n(\theta)| = O\left(\sqrt{C \log(1/\delta)/n}\right)$$

holds for any $\theta$ with probability at least $1 - \delta$, where $C$ is some notion of complexity.

Let $\theta_* = \arg\min_\theta L(\theta)$ and $\hat{\theta} = \arg\min_\theta L_n(\theta)$. From the properties of $\theta_*$ and $\hat{\theta}$, and the triangle inequality, we get that

$$
\begin{aligned}
|L(\hat{\theta}) - L(\theta_*)| &= |L(\hat{\theta}) - L_n(\hat{\theta}) + L_n(\hat{\theta}) - L(\theta_*)| \\
&\leq |L(\hat{\theta}) - L_n(\hat{\theta}) + L_n(\theta_*) - L(\theta_*)| \\
&\leq |L(\hat{\theta}) - L_n(\hat{\theta})| + |L_n(\theta_*) - L(\theta_*)| \\
&= O\left(\sqrt{C\log(1/\delta)/n}\right)
\end{aligned}
$$

holds with probability at least $1 - 2\delta$. Therefore, as $n$ increases, $\hat{\theta}$ performs similarly to $\theta_*$. Now note that $\hat{\theta}$ is the learned quantitative judge parameter. Moreover, the base judge cannot perform better than $\theta_*$ because it can be instantiated using our parameters and $\theta_*$ is the optimal solution. Therefore, as the sample size $n$ increases, the quantitative judge is guaranteed to perform at least as well as its base judge with a high probability.

