# OpenReview forum: "Quantitative LLM Judges"
_TMLR — Decision pending for TMLR_

### Review · Reviewer_pL2q · 2026-06-07

**Summary Of Contributions:**

**Summary**

The paper tackles a real and practical problem: LLM judges tend to produce thoughtful textual rationales but their numeric scores are often poorly calibrated, compressed toward certain values, and misaligned with human ratings. The authors' proposal is refreshingly simple — rather than fine-tuning the entire judge, just train a small regression model on top of the judge's existing output. They freeze the base LLM judge, extract its rationale embedding and score, and learn a generalized linear model (GLM) to predict human-aligned scores from these features. The whole thing requires only 100–500 human labels and a few thousand parameters.

They present four flavors of this idea: a least-squares (LS) judge for regression on absolute scores, a multinomial (MN) judge for classification over Likert-scale categories, a Bradley-Terry-Luce (BTL) judge for pairwise preference using a relative base judge, and a two-headed BTL (BTL2) judge that derives preferences from two separate absolute evaluations. Experiments span four datasets (Summarize from Feedback, HelpSteer2, Offset Bias, Nectar) with comparisons against base judges (Llama-3.1-8B, Prometheus), JudgeLM, Llama-3.1-70B, and TRACT fine-tuning.

The core takeaway is that this lightweight post-hoc calibration consistently beats the base judge on whatever metric it's trained to optimize, often with surprisingly few training examples.

**Strengths**

- The central idea is elegant and well-grounded. Decoupling the "reasoning" role of the LLM from the "scoring" role is a natural move, and the connection to probing literature (Alain & Bengio, 2017; Hewitt & Manning, 2019) gives it solid intellectual footing.
- I appreciate the design choice that lets the model recover the base judge exactly (setting θ = 0_d ⊕ 1, c = 0). This means the quantitative judge starts from a sensible default and can only improve with enough data — at least in theory.
- The parameter efficiency is striking. We're talking about 4k parameters learned from a few hundred examples, compared to 20M+ for TRACT. That's a compelling story for practitioners with limited annotation budgets.
- The BTL2 judge is a genuinely clever design — using the difference of two absolute embeddings for preference prediction. The results on Offset Bias are impressive and support the intuition from Jeong et al. (2024) that pointwise evaluation can be more robust.
- The ablation studies are well-chosen. The regularization sweep (Appendix B.1), the embedding quality degradation experiment (Appendix B.2), and the embedding-vs-score ablation (Table 8) all add real understanding of the method's behavior.
- The confusion matrices in Figures 7–8 do an excellent job of showing *how* the method helps — the base judge never predicts certain scores, and the quantitative judges learn to fill those gaps.
- From a practical standpoint, this is exactly the kind of thing a team would want to try when deploying LLM judges: collect a small set of human ratings for validation, then use them to calibrate the judge. Very actionable.

**Weaknesses**

- The biggest gap for me is the proprietary model claim. The abstract says the framework "can be applied to proprietary models," but every experiment uses open-weight models where you can extract final-layer embeddings. The MiniLM experiment in Appendix B.2 is a reasonable proxy, but it shows degraded results on preference tasks — which actually undermines the claim rather than supporting it.
- Looking at Table 1, the correlation metrics (Pearson, Spearman, Kendall) frequently get *worse* after applying the quantitative judge. In some cases the drop is dramatic — Llama on Summarize from Feedback goes from r=0.340 to r=0.061. That's a nearly complete loss of ranking ability. The paper acknowledges this trade-off but the abstract still says "improve the predictive power" without qualification. A reader who cares about ranking (which is arguably the most common use case) would be misled.
- No standard deviations are reported anywhere despite averaging over 10 random training sets. With n=100 or n=200, I'd expect substantial variance. Without error bars, I genuinely cannot tell whether some of the reported improvements are real or just noise.
- I kept looking for a simple baseline that never appeared: what happens if you just do linear regression (or isotonic regression, or Platt scaling) on the base judge's score alone, without using the embedding at all? Table 8 addresses this partially for BTL2 on one dataset, but that's not enough. For all I know, most of the improvement on absolute tasks comes from learning the score distribution rather than from the embedding features.
- The TRACT comparison feels a bit uneven. TRACT gets 200 training examples while the quantitative judges get up to 500. TRACT still wins on absolute rating tasks. I'd like to see what happens if TRACT also gets 500 — or at least an explanation of why that wasn't done.
- BTL2 requires running the base judge twice (once per response). That's 2x the inference cost of BTL. This is mentioned in passing via Table 4 but never explicitly discussed as a trade-off against BTL2's superior accuracy.
- The theoretical guarantee in Appendix C.2 (quantitative judge is "at least as good" as base judge) assumes no regularization and large n. In practice, regularization is always used and n is small. The theory and practice don't quite match, and the paper doesn't discuss this gap clearly enough.
- On the Nectar dataset, the method underperforms JudgeLM and the 70B baseline for preference prediction, but there's no analysis of *why*. What's different about Nectar? Is the GPT-4-generated ground truth fundamentally different in character? This would help practitioners know when to use the method and when not to.

**Additional Comments:**

- Overall the writing quality is good. The math is clearly presented and the paper structure is logical. I had no trouble following the method descriptions.
- The confusion matrices (Figures 7–8) are probably the most illuminating part of the paper for building intuition. Consider whether one of them could be promoted to the main results section.
- There's a natural connection to reward modeling (especially the Bradley-Terry formulation) that goes largely undiscussed. Some RLHF practitioners might find this paper very relevant but won't find it unless the connection is made explicit.
- The title "Quantitative LLM Judges" is a bit generic. Something like "Post-Hoc Score Calibration for LLM Judges" would communicate the contribution more precisely.
- Small oddity: the paper cites "Wolke & Schwetlick, 1988" for GLMs. The standard reference is Nelder & Wedderburn (1972) or McCullagh & Nelder (1989). The cited paper is about iteratively reweighted least squares, which is a computational method rather than the GLM framework itself.
- I want to reiterate that despite my critical tone, I think the core idea here is good and practically useful. The paper just needs to be more honest about what it has and hasn't demonstrated.

**Audience:**

Yes

**Audience Explanation:**

Absolutely. LLM-as-a-judge is one of the hottest practical problems in the field right now, and the question of how to calibrate judge scores with limited human feedback is directly relevant to a large community. I'd say the interested audience includes:

- Anyone building LLM evaluation pipelines in practice (and that's a lot of people these days)
- Researchers working on human-AI alignment and preference learning
- People studying the representation quality of LLM embeddings
- Practitioners who need cheap, reliable automatic evaluation without the cost of fine-tuning


The contribution is modest - this isn't a paradigm shift - but it's the kind of practical, actionable insight that TMLR is well-suited to publish.

The paper would serve its audience better if the claims were appropriately scoped. Right now, a reader might try this with a proprietary model and be disappointed, or expect correlation metrics to improve on absolute tasks and be surprised when they don't.

**Broader Impact Concerns:**

I don't see major ethical or broader impact concerns with this work. It's a calibration technique applied to existing evaluation systems using publicly available datasets. No new data is collected, no human subjects are involved beyond the pre-existing annotations in the benchmark datasets, and the method doesn't introduce capabilities that could be readily misused.

Two minor notes:
- It might be worth briefly acknowledging that better-calibrated automatic judges could reduce (but not eliminate) the need for human evaluation. There's a risk that improved scores create false confidence and lead teams to skip human review entirely. A sentence in the limitations section would suffice.
- The provenance and annotator conditions for Summarize from Feedback and HelpSteer2 are documented in the original papers but not discussed here. Not critical, but a brief mention would be appropriate.

**Claims And Evidence:**

No

**Claims Explanation:**

I'm recommending "No" here, though I want to be clear this isn't a rejection of the core idea — it's about the gap between claims and evidence.

**What's well-supported:**
- The quantitative judges consistently beat their base judges on whatever metric they're trained to optimize. Tables 1 and 2 show this clearly across all four datasets. The MSE improvements are typically 30%+ and accuracy improvements 20%+. I believe these numbers.
- The parameter efficiency claim (Table 3) is straightforward and convincing — 4k params vs 20M for TRACT.
- BTL2 outperforming BTL is clear from Table 2.
- The embedding importance ablation (Table 8) convincingly shows that the rationale embedding, not the score, drives performance.

**What's not adequately supported:**
- "Our framework can be applied to proprietary models" — this is in the abstract but there's zero experimental evidence with a proprietary model. The MiniLM proxy actually shows degraded preference performance, which if anything suggests caution.
- "Quantitative judges can improve the predictive power of existing judges" — true for optimized metrics, often false for correlation/ranking metrics. The abstract doesn't qualify this. A practitioner reading just the abstract would get the wrong impression.
- "They outperform or are comparable to specialized LLM judges, a larger off-the-shelf model, and fine-tuning by TRACT" — not true uniformly. TRACT wins on absolute tasks. JudgeLM and 70B win on Nectar. The qualifier "in most of our benchmarks" is doing a lot of heavy lifting.
- The "at least as good" guarantee from Appendix C.2 — the theory says one thing (asymptotic, no regularization) and the experiments show another (finite samples, regularized, metrics sometimes worsen).
- Statistical reliability — 10 random runs but no variance reported. Some of the improvements in Table 1 are small enough that they could easily be within noise.

**To change my answer to "Yes," the authors would need to:**
- Report standard deviations and show the improvements are statistically significant
- Either demonstrate the proprietary model claim experimentally or remove it
- Add a score-only calibration baseline to isolate the embedding contribution
- Qualify the improvement claims to specify they apply to optimized metrics

**Requested Changes:**

### [Critical for acceptance] Show us the variance

**Concern:** You averaged over 10 random training sets but never report standard deviations or confidence intervals.

**Why it matters:** With training sets of 100–500 examples, I'd expect non-trivial variance. Some of the improvements in Table 1 are modest enough that they could fall within one standard deviation of the base judge. Without this information, I can't assess which results are reliable and which might be noise.

---

### [Critical for acceptance] Back up or walk back the proprietary model claim

**Concern:** The abstract highlights applicability to proprietary models as a key advantage, but no experiment actually uses one. The proxy experiment (MiniLM embeddings in Appendix B.2) shows degraded preference prediction performance.

**Why it matters:** This claim shapes how readers understand the contribution. If I can't use this with GPT-4 in practice without meaningful performance loss, the framing needs to change.

---

### [Critical for acceptance] Include a score-only calibration baseline

**Concern:** Without a baseline that just recalibrates the base judge's score (simple linear regression on b, isotonic regression, or Platt scaling — without the embedding), I cannot determine how much of the improvement comes from the embedding features vs. simple distribution matching.

**Why it matters:** The confusion matrices (Figures 7–8) show the base judge never predicts certain scores. A simple histogram-matching or affine transform of the score might fix this without needing embeddings at all. If so, the paper's contribution is smaller than presented.

---

### [Important but not necessarily acceptance-critical] Tighten the improvement claims

**Concern:** The abstract says "quantitative judges can improve the predictive power of existing judges" and the conclusion says they "consistently outperform their base judges." But correlation metrics on absolute tasks often get worse — sometimes catastrophically so (r dropping from 0.340 to 0.061).

**Why it matters:** Many practitioners care about ranking utility, not just MSE. The current framing could mislead them.

---

### [Important but not necessarily acceptance-critical] Level the playing field with TRACT

**Concern:** TRACT is trained on 200 examples while your judges get 500. Despite this handicap, TRACT still wins on absolute tasks.

**Why it matters:** The paper positions quantitative judges as competitive with or superior to TRACT, but the comparison isn't quite apples-to-apples.

---

### [Important but not necessarily acceptance-critical] Be upfront about BTL2's cost

**Concern:** BTL2 needs two full base-judge passes (one per response) while BTL needs just one. Table 4 shows the inference times but doesn't explicitly highlight that BTL2 roughly doubles the base cost.

**Why it matters:** A practitioner choosing between BTL and BTL2 needs to know the cost-accuracy trade-off.

---

### [Minor / presentation] Clarify what "black-box" actually means here

**Concern:** There's a tension between saying the base judge is treated as a "black box" and the fact that you need its final-layer embeddings. Those require model weights, which isn't truly black-box.

**Why it matters:** Readers will be confused about what access level is actually required.

---

### [Minor / presentation] Discuss why Nectar is harder

**Concern:** The method underperforms the 70B baseline and JudgeLM on Nectar by ~10-15%, but there's no discussion of why.

**Why it matters:** Helps practitioners understand when to trust the method and when not to.

---

### [Minor / presentation] Dense tables

**Concern:** Tables 1 and 2 pack a lot of information and the gray highlighting that distinguishes optimized vs. non-optimized metrics may not render well everywhere.

**Why it matters:** Readability — the key story can get lost in the numbers.

---

> ### Author Response · Authors · 2026-06-29
>
> Dear reviewer,
>
> Thank you for the detailed review and even more detailed guidance on how to address your comments. We updated the paper and marked all changes in blue. We addressed your comments as follows:
>
> ***[Critical for acceptance]** Show us the variance*
>
> Tables 1 and 2: We added standard errors for all methods that train on randomized training sets. We did not add them for other methods because the test sets are fixed. We state it in Section 5.2.
>
> ***[Critical for acceptance]** Back up or walk back the proprietary model claim*
>
> Sections 1 and 6: We removed the proprietary model claims because they were not supported by experiments.
>
> ***[Critical for acceptance]** Include a score-only calibration baseline*
>
> The experiments in Appendix B.3 already showed a score-only baseline for BTL2 on Offset Bias dataset. We expanded these experiments.
>
> ***[Important but not critical]** Tighten the improvement claims*
>
> Abstract, and Sections 1 and 6: We revised the claims. We claim improvements over base judges on optimized metrics and being competitive with our baselines.
>
> ***[Important but not critical]** Level the playing field with TRACT*
>
> We justify TRACT's sample size of 200 in Section 5.2 and explain how we compare it fairly to quantitative judges in Section 5.3.
>
> ***[Important but not critical]** Be upfront about BTL2's cost*
>
> We state it in Section 4.4.
>
> ***[Minor / presentation]** Clarify what "black-box" actually means here*
>
> Section 4: We clarify that the embedding can be obtained from the base judge when it is open-weight, as described in Section 5.2, or from another model, as in Appendix B.2.
>
> ***[Minor / presentation]** Dense tables*
>
> Tables 1 and 2 changed.
>
> ***[Not well supported]** Theory says one thing (asymptotic, no regularization) and the experiments show another (finite samples, regularized, metrics sometimes worsen)*
>
> Sections 4.1 and 6: We call out theory limitations.

---

### Review · Reviewer_XTLz · 2026-06-16

**Summary Of Contributions:**

The paper addresses a known weakness of "LLM-as-a-judge" setups, i.e when one LLM grades another LLM's output, it writes good qualitative reasoning but assigns poor numeric scores. They don't match human ratings and tend to be mis-calibrated.

The authors' fix is to leave the LLM judge untouched and bolt a small, classic model on top of it. The base judge produces its usual rationale and rough score; a lightweight generalized linear model then reads an embedding of that rationale, together with the judge's own score, and predicts the score a human would give. This add-on is trained on only 100–500 human-labeled examples. Because the base judge is treated as a black box, the method also works with proprietary models that can't be fine-tuned, and the add-on is deliberately built so that, in the worst case, it just reproduces the base judge's score (so it shouldn't do worse than the starting point).

They provide four versions of this add-on for four grading situations: a Least-Squares (LS) judge for scoring a single answer on a continuous scale, a Multinomial (MN) judge for scoring a single answer on a category scale (e.g., 1–5 Likert), a Bradley-Terry-Luce (BTL) judge for picking the better of two answers, and a two-headed BTL (BTL2) judge that scores each answer separately and then compares. They evaluate on two human-rated datasets (Summarize from Feedback, HelpSteer2) and two synthetic/relative datasets (Offset Bias, Nectar), comparing against the base judges, specialized judges (Prometheus, JudgeLM), a much larger off-the-shelf judge (Llama-3.1-70B), and fine-tuning (TRACT), with ablations on regularization, embedding quality, and the relative contribution of the rationale vs. the score.

Strengths:

- The idea is simple, cheap, and practical: a few-thousand-parameter model trained on a few hundred labels, versus 20M+ parameters for fine-tuning.
- It is general - one framework spans absolute and relative grading, regression and classification, and works on black-box judges.
- The "no worse than the base judge" property is a clean design choice, backed by an argument in Appendix C.2.
- Evaluation is broad (multiple datasets, two base judges, several baseline types, three sample sizes) and the paper is mostly candid about where it loses.


Weaknesses

- Each version only improves the one metric it optimizes, and often drags the others below the base judge. The headline claims don't carry this qualifier.
- Relatedly, the regression version lowers its error mainly by predicting close to the average score, which is the same flattening behavior the paper criticizes in the intro. So a better error number doesn't mean a more useful judge, and that should be said plainly.
- A couple of "comparable to base" lines are too kind. On one preference dataset with one base judge, the method is actually a bit worse across the board.
- The "aligns to humans" framing leans on two datasets whose ground truth isn't human - one is synthetic and the other is labeled by GPT-4.
An ablation shows a generic off-the-shelf text encoder matches the base judge's own representation on the rating tasks, which undercuts the motivation for using the judge's representation in that setting.
- No error bars are reported despite averaging over ten runs. The big results don't need them, but the close cross-method comparisons and the smallest-sample cells do.

**Audience:**

Yes

**Audience Explanation:**

LLM-as-a-judge is everywhere right now and miscalibrated judge scores are a real, unsolved annoyance. A method that fixes a black-box judge's numbers after the fact from a few hundred labels and a few thousand parameters, no weight access, no fine-tuning, etc is going to interest both people building eval pipelines and people studying judge calibration. The finding that a tiny GLM on frozen rationale embeddings can match or beat much bigger or fine-tuned judges on the targeted metric and the ablation showing the rationale matters far more than the score (Appendix B.3), are each worth knowing on their own.

**Broader Impact Concerns:**

Nothing that needs a full Broader Impact Statement, in my view; at most a short note. This is a calibration method built on public datasets; it doesn't add generative capability or any obvious risk. The one thing worth a sentence is that calibrating to 100–500 labels in one domain can create a false sense that the judge is "aligned" more generally - a judge tuned to match a narrow human sample still inherits that sample's biases, and shouldn't be read as broadly fair or robust. A line making that explicit would be enough.

**Claims And Evidence:**

Yes

**Claims Explanation:**

The core result is convincing. I'd recommend some slight wording changes. The paper says the judges "consistently outperform" the base, but that's only true for the one metric each version targets - they often get worse on the others, which the paper admits in the results but not in its headline claims. The regression version in particular lowers its error mainly by guessing close to the average score, which is the same flattening behavior the paper criticizes in the intro, so a lower error number shouldn't be read as a genuinely better judge.

**Requested Changes:**

Calibrate the headline claims. Change "consistently outperform" to "outperform on the optimized metric", and note that the other metrics often get worse.

Fix the one preference-dataset description. Call that result an underperformance versus the base judge, not "comparable," and square it with the broader "generally outperforms" line.

Scope the human-alignment claim. State plainly that one preference dataset is synthetic and the other is labeled by GPT-4, and limit "alignment to human scores" to the genuinely human-rated datasets.

Would strengthen it, but not required:

Add error bars where the comparisons are close - the cross-method ones and the smallest-sample rows. You already ran ten seeds, so reporting the spread is nearly free, and it's what's needed to defend the "comparable" and "slightly worse" lines.

Specify how the rationale representation is extracted from the judge, and say whether code will be released.

Address the generic-encoder finding head-on: it matches the judge's own representation on rating tasks, so frame the judge-representation advantage as specific to preference tasks.

---

> ### Author Response · Authors · 2026-06-29
>
> Dear reviewer,
>
> Thank you for the detailed and supportive review, and noting that our approach has many properties that make it desirable in practice. We updated the paper and marked all changes in blue. We addressed your comments as follows:
>
> ***[Critical for acceptance]** Calibrate the headline claims. Change "consistently outperform" to "outperform on the optimized metric", and note that the other metrics often get worse.*
>
> Abstract, and Sections 1 and 6: We revised the claims. We claim improvements over base judges on optimized metrics and being competitive with our baselines.
>
> ***[Critical for acceptance]** Fix the one preference-dataset description. Call that result an underperformance versus the base judge, not "comparable," and square it with the broader "generally outperforms" line.*
>
> Section 5.3: We clearly call out underperformance.
>
> ***[Critical for acceptance]** Scope the human-alignment claim. State plainly that one preference dataset is synthetic and the other is labeled by GPT-4, and limit "alignment to human scores" to the genuinely human-rated datasets.*
>
> Section 5.5: We merged the paragraph with confusion matrices (originally in Appendix B.4). We do not talk about human alignment when discussing experiments on synthetic datasets.
>
> ***[Would strengthen work but not required]** Add error bars where the comparisons are close - the cross-method ones and the smallest-sample rows. You already ran ten seeds, so reporting the spread is nearly free, and it's what's needed to defend the "comparable" and "slightly worse" lines.*
>
> Tables 1 and 2: We added standard errors for all methods that train on randomized training sets. We did not add them for other methods because the test sets are fixed. We state it in Section 5.2.
>
> ***[Would strengthen work but not required]** Specify how the rationale representation is extracted from the judge, and say whether code will be released.*
>
> We added this to Section 5.2. The code will be released.
>
> ***[Would strengthen work but not required]** Address the generic-encoder finding head-on: it matches the judge's own representation on rating tasks, so frame the judge-representation advantage as specific to preference tasks.*
>
> We are not completely sure what this suggestion was. We added to Section 5.2 that the rationale embedding can be computed by another model.

---

### Review · Reviewer_3B4u · 2026-06-17

**Summary Of Contributions:**

The paper proposes a lightweight framework for calibrating LLM judges to better align with human feedback in any domain of choice. The core methodology is based on the fact that LLM are good are qualitative evaluation but their quantitative evaluations are not calibrated. The method uses the rationale and score of an LLM judge to train a lightweight GLM model on a small amount of human labeled data to predict calibrated scores. The paper studies 2 variants each for absolute(LS and MN) and relative feedback(BTL1 and BTL2) and evaluates their capabilities across different datasets. The main strengths of the paper is the simplicity of the approach, light weight computationally efficient methodology, and informative ablation studies. However, some of the claims like quantitative judges “consistently outperform their base judges” is overstated. While the proposed models improve the metrics that they directly optimize, they do not consistently outperform the base judges across correlation metrics particularly in the case of the absolute judges.

**Audience:**

Yes

**Audience Explanation:**

The problem of improving the alignment and calibration of LLM judges is important and timely. The paper proposes a simple, computationally efficient approach that is likely to be of interest to researchers working on LLM evaluation.

**Broader Impact Concerns:**

No concerns.

**Claims And Evidence:**

No

**Claims Explanation:**

1. The claim that the proposed quantitative LLM judges consistently outperform their base judges is not fully supported. In the absolute rating setting, LS and MN improve MSE, MAE, or accuracy, but they sometimes degrade correlation metrics relative to the base judges. A related concern is that, based on Figure 7 the test set appears highly skewed toward higher scores. A model that predicts the majority class or the mean score could give high performance. So metrics such as accuracy, MSE and MAE may be insufficient by themselves.
2.The claim that the method applies naturally to proprietary black box judges is not convincingly demonstrated. The proposed method uses embeddings from the base judge which are not available from proprietary models.
3. In the case of relative feedback the paper evaluates on Offset Bias and Nectar dataset, but these are synthetic and GPT-4 derived preference datasets rather than direct human preference datasets. Therefore, these results do not fully support broad claims about alignment to human preferences in relative evaluation.

**Requested Changes:**

Critical to acceptance:
1. Review the claim that quantitative judges consistently outperform their base judges. The current evidence shows improvements mainly on the metrics directly optimized by the proposed models, while correlation metrics often degrade in the absolute-rating setting. The paper should revise this claim and discuss the trade-off between score prediction metrics and correlation metrics more explicitly.
2. Revise or supplement the metrics in Table 1 to better handle imbalanced score distributions. Since the absolute rating datasets appear skewed toward higher scores, accuracy, MSE, and MAE may be insufficient by themselves. The paper should include more imbalance aware metrics.
3. Clarify or better support the proprietary model claim. The method relies on embeddings from the base judge which are not generally available for proprietary models. The authors should either weaken this claim or add experiments using generated rationales and scores from proprietary models with open source text embeddings.

Would strengthen the work:
1. Provide uncertainty estimates and statistical testing. Since the method is trained on small labeled subsets and some gains are modest or non-monotonic, the paper should report standard deviations, confidence intervals, or significance tests across random training subsets and test examples.
2. Better support or narrow the claims about alignment with human preferences. The relative feedback experiments use synthetic datasets, which do not fully support broad claims about alignment with human preferences. The paper should include at least one human preference benchmark for relative evaluation or discuss this clearly in the limitations.

---

> ### Author Response · Authors · 2026-06-29
>
> Dear reviewer,
>
> Thank you for the detailed review, and recognizing that our simple and computationally-efficient approach could be of a broad interest. We updated the paper and marked all changes in blue. We addressed your comments as follows:
>
> ***[Critical for acceptance]** Review the claim that quantitative judges consistently outperform their base judges.*
>
> Abstract, and Sections 1 and 6: We revised the claims. We claim improvements over base judges on optimized metrics and being competitive with our baselines.
>
> ***[Critical for acceptance]** Revise or supplement the metrics in Table 1 to better handle imbalanced score distributions.*
>
> The opposite is true: the absolute rating distributions are not heavily skewed towards a single value. We state it in Section 5.1.
>
> ***[Critical for acceptance]** Clarify or better support the proprietary model claim.*
>
> Sections 1 and 6: We removed the proprietary model claims because they were not supported by experiments.
>
> ***[Important but not critical]** Provide uncertainty estimates and statistical testing.*
>
> Tables 1 and 2: We added standard errors for all methods that train on randomized training sets. We did not add them for other methods because the test sets are fixed. We state it in Section 5.2.
>
> ***[Important but not critical]** Better support or narrow the claims about alignment with human preferences.*
>
> Section 5.5: We merged the paragraph with confusion matrices (originally in Appendix B.4). We do not talk about human alignment when discussing experiments on synthetic datasets.